# Allocating capital-associated CO$_2$ emissions along the full lifespan of capital investments helps diffuse emission responsibility

Quanliang Ye [1,2,3], Maarten S. Krol[1], Yuli Shan [4] ✉, Joep F. Schyns [1], Markus Berger[1] & Klaus Hubacek [2] ✉

Capital assets such as machinery and infrastructure contribute substantially to CO$_2$ emissions over their lifetime. Unique features of capital assets such as their long durability complicate the assignment of capital-associated CO$_2$ emissions to final beneficiaries. Whereas conventional approaches allocate emissions required to produce capital assets to the year of formation, we propose an alternative perspective through allocating required emissions from the production of assets over their entire lifespans. We show that allocating CO$_2$ emissions embodied in capital assets over time relieves emission responsibility for the year of formation, with 25–46% reductions from conventional emission accounts. This temporal allocation, although virtual, is important for assessing the equity of CO$_2$ emissions across generations due to the inertia of capital assets. To re-allocate emission responsibilities to the future, we design three capital investment scenarios with different investment purposes until 2030. Overall, the existing capital in 2017 will still carry approximately 10% responsibilities of China's CO$_2$ emissions in 2030, and could reach more than 40% for capital-intensive service sectors.

Substantial government and private investments in capital assets such as power plants, machinery, and infrastructure have enabled global fast-growing economic activities[1,2]. Capital investments account for around one quarter of global gross domestic product (GDP) since 1970[3]. In some developing countries, for instance China, capital investments could account for up to 47% of its national GDP, with an annual average growth rate of 12% since 1995[3]. Building up capital assets requires considerable resource inputs and generates pollution[4]. For instance, 156 gigatons (Gt) of carbon dioxide (CO$_2$) have been emitted to produce global capital assets invested between 1995 and 2015, accounting for 32% of global total CO$_2$ emissions during the same period[5].

Two unique features of capital assets, different from non-capital products, complicate the assignment of capital-associated CO$_2$ emissions to final capital beneficiaries, and thus require capital-oriented methods (as complements of conventional production-[6,7] and consumption-based approaches[8,9]) for the assignment. First, capital assets are invested and used by economic sectors for productive purposes. Between the initial investment and production-oriented use phase, capital assets are produced by capital-producing (so-called 'capital formation' in national accounting) sectors. This feature raises arguments about how to allocate CO$_2$-emission responsibilities of capital activities[10–14], to producers, or to users, or to final consumers of products that are produced by using associated assets. Second, capital assets can exist for several years or even decades, and serve economic production throughout their lifespans. This feature implies that future production and consumption will induce not only economic inputs and emissions in the future, but also depend on already-existing capital

[1]Multidisciplinary Water Management, Faculty of Engineering Technology, University of Twente, 7522 NB Enschede, the Netherlands. [2]Integrated Research on Energy, Environment and Society (IREES), Energy and Sustainability Research Institute Groningen (ESRIG), University of Groningen, 9747 AG Groningen, the Netherlands. [3]Department of Planning, Aalborg University, 9000 Aalborg, Denmark. [4]School of Geography, Earth and Environmental Sciences, University of Birmingham, Birmingham B15 2TT, UK. ✉e-mail: y.shan@bham.ac.uk; k.hubacek@rug.nl

assets and embodied emissions that occurred in the past as long as the capital is used. It hence leads to the temporal allocation of emission responsibilities of capital activities along capital's full lifespan.

Little is known about the second feature of capital assets and its impacts on the allocation of environmental pressures along capital's lifespan. Literature has investigated the geospatial displacement of environmental pressures along supply chains, and allocated them from producers to final consumers, yielding consumption-based environmental pressures[8,9]. Due to aforementioned two features of capital, how to treat the purchase of capital assets and allocate associated environmental responsibilities is still debated. In conventional analysis, capital assets are treated in the same way as products for final consumption, allocating environmental pressures that occurred during the production of capital assets to the purchasing sectors and countries[4,8,9]. Recently, studies treated capital as one production factor (i.e., as intermediate inputs for economic production), and allocated capital-associated environmental pressures to final consumption across sectors and countries[12–14]. Particularly, Chen et al.[11] considered the dynamic of capital investment and depreciation in national greenhouse gas (GHG) emission assessment for a single year 2009. However, the intertemporal features of capital assets remain unaddressed (see the comparison of existing capital-oriented methods in Supplementary Information Table S1), neglecting that capital assets used for year $n$'s production are from different time cohorts–produced based on different production recipes, trade networks, and environmental intensities. This neglect has been found to result in an approximate 30% underestimation in capital-associated GHG emissions[12].

To properly understand this important temporal dimension of environmental responsibility displacement requires a full picture of capital flows across sectors and regions (according to the first feature) and also throughout its lifespan from the past to the future (according to the second feature). This study presents a comprehensive analysis of capital development and quantifies temporal $CO_2$ displacement along capital production, trade and use over the period of 1995−2017 as well as under capital investment scenarios for the near future until 2030. We first develop an inter-provincial capital-endogenized multi-regional input-output (MRIO) model to link provincial capital depreciation to the production side of capital using sectors, and subsequently to the consumption side of final products of each province (Methods). Second, to understand the temporal dimension of $CO_2$ displacement from the past to the future, we design China's future capital investment pathways by a 'business-as-usual' (BAU) scenario and two alternative capital-oriented scenarios until 2030 (Methods). One of the capital-oriented investment pathways is developed on the principle of improving economic growth and social well-being (KES, here 'K' standing for capital), under which China is increasingly focusing on the role of capital assets, especially infrastructure related to transportation and communications. The other pathway is based on the principle of low-carbon development (KLC), under which China's future capital investment are required to focus on low-carbon technologies by the electricity generation sector and end-use sectors such as transportation services. We quantify spatiotemporal $CO_2$ displacement embodied in capital flows among sectors, across provinces, and over time. We show that temporally displaced $CO_2$ emissions along capital's full lifespan take a large share in total emissions of China, and capital-intensive service sectors. This temporal displacement, although virtual, is also important for assessing the sustainability and efficiency of national resource use especially in developing countries, which may have capital investment booms during short periods, and the equity of resource use across generations.

## Results

### Monetary capital flows and embodied $CO_2$ emissions over time

Distinguishing capital formation from capital investment and use is a prerequisite to understand the full lifespan of capital. Monetary capital flows (Fig. 1a) start from capital-investing sectors (i.e., sectors

undertaking the investment to build up their capital stock) to capital-producing (i.e., capital formation) sectors, and end in capital-using sectors (i.e., the original investing sectors) for production to satisfy final demand. Real estate services, transportation services, electricity generation, and residential services are the main capital-investing sectors in China, which together accounted for half of total capital investment during the period of 1995−2017. Information of annual capital formation is recorded as gross fixed capital formation (GFCF) in the national accounts, which show that construction (contributing 58% of total GFCF), general equipment manufacturing (7%), and transportation equipment manufacturing (5%) dominated China's capital formation over the time period 1995−2017. As such, main flows from capital investment to formation are observed among these key capital investing and producing sectors. Capital-using sectors will take over capital assets produced by capital-formation sectors for their productive purposes over years. We find that approximately one-third of all the capital assets formed during 1995−2017 have been depreciated to produce final consumption (14%), fixed capital (12%), and international exports (6%) by 2017. The remaining assets are still effective for future economic activities. Based on the three capital scenarios developed in this study, we show that another third (31%) of the capital assets formed in 1995−2017 would be depreciated between 2018 and 2030.

Our study reveals that conventional estimations of supply chain-wide $CO_2$ emissions (i.e., consumption-based $CO_2$ emissions, CBE) embodied in 'capital investment' are, to some extent, misleading the allocation of capital-associated emission responsibilities to capital producers instead of capital users. Sectors that mainly contribute capital-associated emission flows are the same as those in the monetary flows (Fig. 1b). That is, the construction sector took the largest share in CBE of capital formation ($CBE^{GFCF}$, Methods), accounting for 68% of total $CBE^{GFCF}$ during 1995−2017. This result is consistent with previous findings regarding CBE of GFCF sectors[9,15]. As for supply chain-wide $CO_2$ emissions embodied in depreciated capital ($F^K$, Methods), real estate services, transportation services, and residential services are the main contributors. $F^K$ can also be allocated to final goods and services until 2017, between 2018 to 2030, and for long-term future production after 2030, which account for 35%, 33%, and 32%, respectively, of total $CBE^{GFCF}$ during 1995−2017. It is important to note that supply chain-wide $CO_2$ emissions embodied in capital investment and use have rarely been estimated for actual investing sectors. Previous supply chain-wide $CO_2$ emissions of 'capital investment' were calculated for capital formation sectors, i.e., $CBE^{GFCF}$[9,15]. As shown in Fig. 1a, capital investing sectors——the final users of capital assets——are different from capital formation sectors. Therefore, a misallocation of capital-associated emission responsibilities to capital producers instead of capital consumers is revealed in conventional input-output table-based estimates of supply chain-wide $CO_2$ emissions of 'capital investment'. When re-allocating this part of capital-associated $CO_2$ emissions to the actual capital consumers or further to final goods and services throughout the full lifespan of capital, it would substantially alter $CO_2$ emission accounting at both regional and sectoral levels.

### Capital re-allocation altering regional $CO_2$ emission accounts

How we assign capital-associated $CO_2$ emissions substantially alters regional $CO_2$ emission accounts from both production and consumption perspectives (Fig. 2). Conventionally, we treat capital assets the same way as non-capital goods, and assign capital-associated $CO_2$ emissions to the producing region yielding production-based $CO_2$ emissions (PBE) of GFCF or to the purchasing region yielding CBE of GFCF. In this study, we treat capital assets as production inputs by endogenizing capital investment and consumption into economic production over time and across provinces (Methods), and re-allocate supply chain-wide $CO_2$ emissions embodied in annual capital

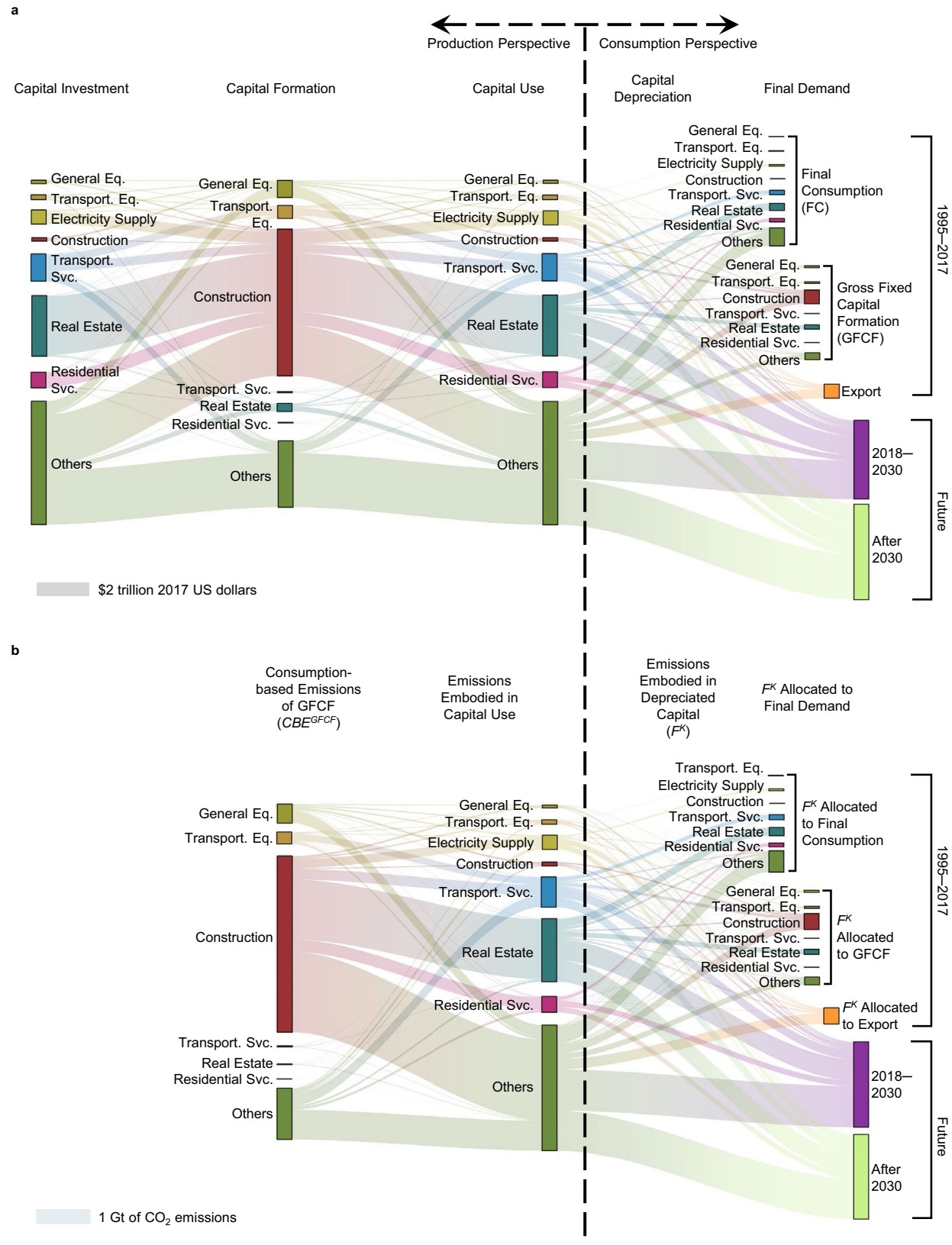

depreciation ($F^k$) to capital using sectors for production-based accounting or to final demand for consumption-based accounting.

National PBE and CBE after $F^k$ re-allocation (denoted as PBE$^k$ and CBE$^k$) are lower than conventional PBE and CBE (Fig. 2a). The reason is that only one-third of the $CO_2$ emissions embodied in GFCF occurring during 1995–2017 would be assigned to economic production over the

same period (Fig. 1b). From a production perspective, compared with conventional PBE, national PBE$^k$ would be 25–35% lower since 1995. The decrease in national PBE implies that conventional PBE of GFCF in a certain year is still larger than cumulative $F^k$ from 1995 to that year for production of capital using sectors. The changes would even be larger from a consumption perspective with 31–46% decrease from

**Fig. 1 | Capital and CO$_2$ flows along economic activities and time. a** Monetary capital flows across key sectors of China's capital development. An example to understand different terms of capital in (**a**): the transportation services sector investing in transportation equipment (e.g., vehicles) is regarded as 'capital investment'; the manufacturing sector producing vehicles is regarded as 'capital formation'; the transportation services sector using vehicles is regarded as 'capital use'; during the use of vehicles, the annual decline of the total value of vehicles is regarded as 'capital depreciation'. A time difference between capital investment and use should be also noted. Compared to equipment like vehicles, this time difference is more critical for infrastructure and buildings that takes years to be built before being used by the investors. **b** Embodied CO$_2$ transfers across key sectors of China's capital development. The cumulative amounts of capital investment and embodied CO$_2$ emissions between 1995 and 2017 are shown in the plots. Capital-related flows for the period 2018–2030 are shown as average flows of all three scenarios developed in this study. Seven key sectors highly relevant for China's capital development are selected in the plots. Full names of sectors can be found in Table S6.

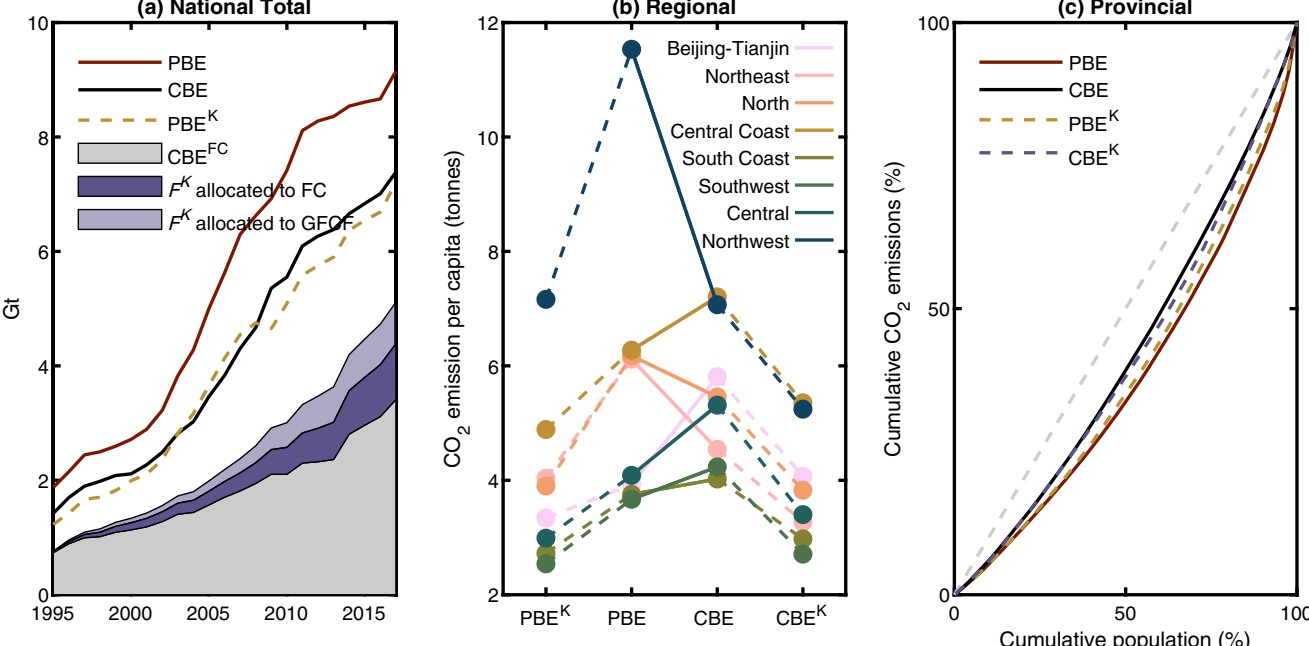

**Fig. 2 | Alteration to production-based emissions (PBE) and consumption-based emissions (CBE) due to capital re-allocation. a** National PBE with (PBE$^K$, represented by the orange dashed line) and without (conventional PBE, represented by the red solid line) the re-allocation of capital-associated CO$_2$ emissions ($F^K$), as well as the national CBE with (stacked grey, purple and light purple areas) and without (i.e., the conventional CBE, represented by the black solid line) the re-allocation of $F^K$. **b** Changes in regional per-capita PBE and CBE for the year 2017 with and without the re-allocation of $F^K$. The geographical partition of China can be found in Table S4. **c** Inter-provincial inequality of per-capita PBE and CBE in 2017 with and without the re-allocation of $F^K$. The gray dashed line represents the perfect equality of per-capita CO$_2$ emissions.

conventional CBE. We also observe that the relative changes in recent years from conventional emission accounts to our capital-endogenized accounting method are generally smaller than those in the early years around 1995. Economic growth needing more capital inputs is one reason, whilst neglecting pre-1995 capital investment and their CO$_2$ emissions (due to lacking data) for current production is another. Our estimates of capital consumption and embodied $F^K$ are conservative, especially for the early years in our modeling period (e.g., year 1995). For later years, the impacts of neglecting pre-1995 capital investments become much smaller. Starting with 2013, less than 1% of the $F^K$ was allocated from CO$_2$ emissions embodied in capital goods invested in 1995.

Changes are also observed in per-capita CO$_2$ emissions and inter-provincial inequality due to re-allocating capital-associated CO$_2$ emissions. Similar to changes in national PBE and CBE, regional per-capita PBE$^K$ and CBE$^K$ also decline compared with conventionally accounted emissions (Fig. 2b). Changes in per-capita CO$_2$ emissions vary widely among regions for the year 2017, especially per-capita PBE. The per-capita PBE$^K$ are observed with a range of 15–38% reduction from the conventional per-capita PBE in 2017 (detailed percentage changes are listed in Table S9). The Northwest, the North, and the Northeast have relatively larger reduction in their regional per-capita PBE, compared with more developed regions such as Beijing-Tianjin

and the Central Coast. Yet, these northern regions only invested around 28% of the total capital formation in 2017. We also observe that changes in per-capita PBE to PBE$^K$ are consistent with regional changes from conventional per-capita PBE to CBE. Thus, to explain the relatively larger changes in per-capita PBE in these northern regions, their net-exporting roles of capital assets and embodied CO$_2$ emissions may be the main reason. In contrast, the relatively larger changes in per-capita CBE are found in the regions having more capital investment, such as the Central (−36%) and the Southwest (−36%) which contributed 28 and 17%, respectively, of national total GFCF in 2017. Lastly, our results also reveal that capital re-allocation would decrease inter-provincial inequality of per-capita PBE, but also to some extent increase the inequality of per-capita CBE (Fig. 2c).

## Conventional PBE and CBE under different capital-investment scenarios

To reveal the full lifespans of capital assets from the past to the future, future production and use of capital in China are projected by a 'business-as-usual' (BAU) scenario and two capital-oriented pathways (i.e., KES and KLC scenarios) that focus on different purposes of capital investment (Methods). Conventional PBE and CBE of China would substantially increase under the BAU and KES scenarios (Fig. 3a) but would only show modest growth (less than 2%) under the KLC

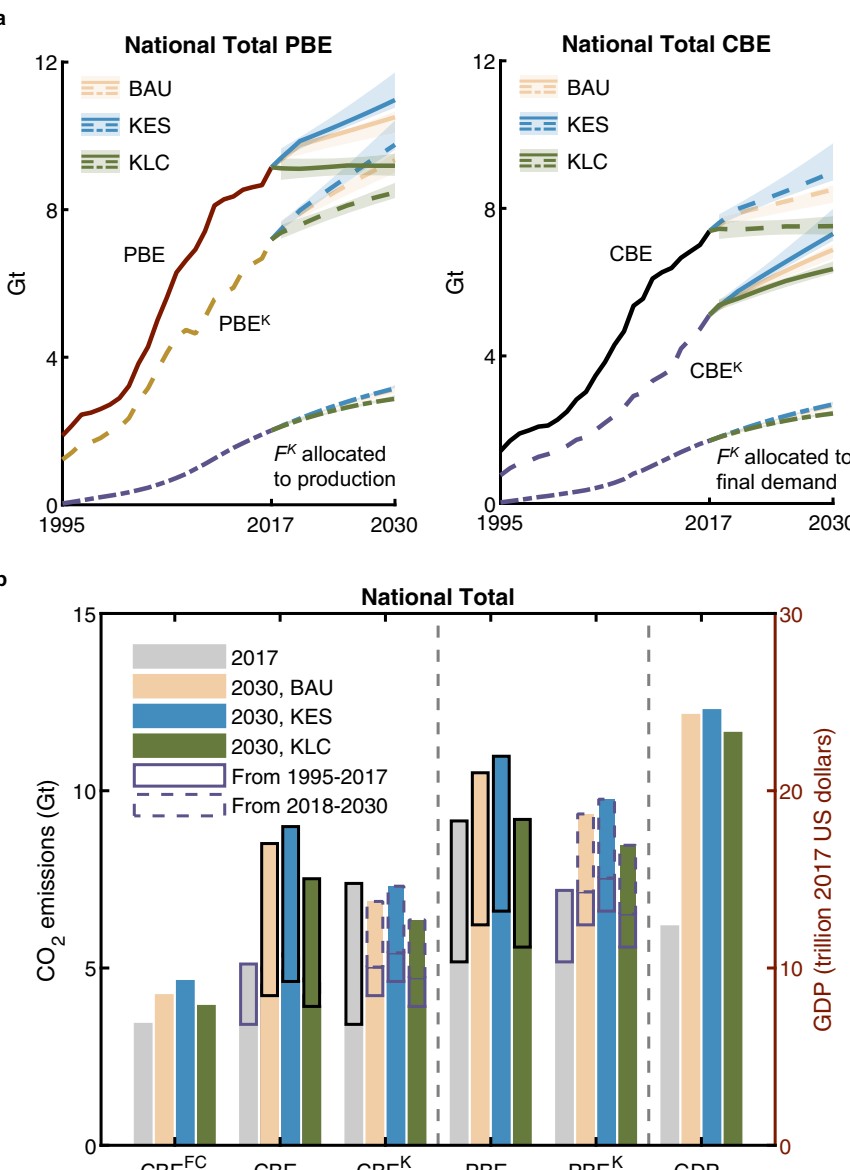

**Fig. 3 | National CO$_2$ emissions with and without re-allocation of capital-associated CO$_2$ emissions under the 'business-as-usual' (BAU), capital for economy and social well-being (KES), and capital for low-carbon development (KLC) scenarios until 2030. a** National CO$_2$ emissions of China under different scenarios. Colored areas represent the 25th–75th percentile of the results of the uncertainty analysis. **b** National CO$_2$ emissions and gross domestic product (GDP) for 2017 and 2030 under different scenarios. Re-allocated CO$_2$ emissions embodied in capital depreciation ($F^k$) are disaggregated into those that occurred in the period 1995−2017 (with solid purple edge line) and those that would occur in the period 2018−2030 (with dashed purple edge line).

scenario. The national PBE in 2030 (Fig. 3b) would increase by 15% under the BAU scenario from the base-year level, and by 20% under the KES scenario because more investment will be made in infrastructure for economic growth and improvement of social well-being. The main growth in national PBE under the BAU and KLC would be observed in material manufacturing sectors (Fig. S3) which provides materials such as cements for infrastructure construction and metal products for machinery production. The growth of national PBE would also be offset by CO$_2$ emissions from electricity generation (Figure S3) given the efficiency improvement and mix changes in energy production and use. From the consumption perspective, similar growth rates would also be found in national CBE of final consumption and final demand (Fig. 3b) while the main growth is observed in the construction sector due to the formation of fixed capital and transportation services due to increasing demand (Fig. S7). Uncertainty analysis shows that national CO$_2$ emissions in 2030 would have the largest fluctuation of −4–6% under the KES scenario (Fig. 3a). Moreover, compared with the BAU

scenario, an extra 7% of investment in low-carbon technology under the KLC scenario would gain a 9% reduction in national PBE, but would also result in a 4%-decrease in national GDP. In addition, at the regional level, potential decreases in provincial PBE and CBE in 2030 could be expected in some regions such as Beijing-Tianjin, and the Southwest (Fig. S2). Detailed analysis of future projections of conventional PBE and CBE of each region can be found in Supplementary Information 3.

**'Historically committed' CO$_2$ emissions for future production and consumption**

When continuing allocating $F^k$ that occurred during the period 1995−2017 to economic production and consumption in the near future, our results show that approximately 10% of national CO$_2$ emissions (represented by PBE$^k$ or CBE$^k$) in 2030 would be based on CO$_2$ embodied in capital investments during 1995−2017 (Fig. 3b). This share of pre-2017 $F^k$ in national CO$_2$ emissions would be even higher in 2018 and 2019, accounting for 23−30%, since the re-allocated $F^k$ from a

certain year decreases along the lifespans of assets (Fig. S4). The total share of $F^K$ (including both pre-2017 emitted and future emissions between 2018 and 2030) in national $PBE^K$ and $CBE^K$ would be respectively 32−34% and 37−39% in 2030, and both have ±2% fluctuations based on the uncertainty analysis.

We regard this part of capital-associated emissions (i.e., re-allocated $F^K$) that occur before a certain year $n$ but are finally allocated to economic production and consumption in year $n$ as 'historically committed' $CO_2$ emissions. The 'historically committed' $CO_2$ emissions have some differences from future 'committed' $CO_2$ emissions that were estimated by Davis et al.[16] and Tong et al.[17]. The future 'committed' $CO_2$ emissions look forward at the expected $CO_2$ emissions after year $n$ by operating existing fossil fuel-burning infrastructure by year $n$. Our 'historically committed' $CO_2$ emissions look backward at how much of the historically emitted $CO_2$ that was embodied in existing capital assets by year $n$ should be attributed to economic production and consumption in year $n$, as they rely on these histori-cally formed capital assets. Based on the three capital scenarios, this study extends the analysis of spatiotemporal downstream impacts of capital development on regional $CO_2$ emission accounting, and indeed quantify both historically and future 'committed' $CO_2$ emissions of all economic sectors, while the previous future 'committed' $CO_2$ emis-sions were only estimated for the power generation sector[16,17].

Contributions of historically and future 'committed' $CO_2$ emis-sions to sectoral $CO_2$ emission (represented by $PBE^K$ or $CBE^K$) vary widely in our analysis. Table 1 summarizes historically and future 'committed' $CO_2$ emissions of four capital-intensive production sec-tors in 2030 under each scenario. We find that most $CO_2$ emissions of the electricity generation and supply sector are future 'committed' emissions, as highlighted in previous studies[16,17], whereas 'historically committed' $CO_2$ emissions of its production and consumption are relatively small (only accounting for 4−6%). In contrast, 'historically committed' $CO_2$ emissions of service-related sectors would occupy a large share of their future $CO_2$ emissions. Particularly for real estate services and residential services, 'historically committed' $CO_2$ emissions would dominate their future $CO_2$ emissions from both production and consumption perspectives, accounting for more than 83% of their $CO_2$ emissions. The remaining emissions are attributable to economic activities of real estate services in the year 2030. This figure is in line with the 77%-86% range of changes for carbon footprints of residential housing, found by Berrill et al.[18], based on a capital-endogenized model of the United States. Transportation services would have less differ-ence between its historically and future 'committed' $CO_2$ emissions, compared with other sectors, and would have more future 'committed' emissions (contributing more than 60%) in 2030. Furthermore, 'his-torically committed' $CO_2$ emissions from 1995−2017 would take the largest share in $CO_2$ emissions of China and most sectors in 2030 under the KLC scenarios. This is because cleaner production under the KLC scenario would reduce associated $CO_2$ emissions of production and consumption in future, which hence enlarges the share of histor-ical $CO_2$ emission embodied in capital production that relied on lower-efficient production technologies. Our results suggest that the earlier development of efficient productive capital would bring less $CO_2$ emissions of future production, as observed in developed countries such as the United States and Japan[15,19].

## Discussion

The two features of capital assets raise two important topics of ana-lyzing capital activities and their environmental responsibilities. One is the allocation of environmental responsibilities across different capital activities such as capital formation or capital use, the other is temporal displacement of environmental responsibilities along capital's life-span. The first topic has been explored by endogenizing capital into MRIO modelling[10–14], while the second topic has not received much attention. This study explores the second issue and demonstrates an

**Table 1 | Sectoral $CO_2$ emissions (in million tonnes) for the year 2030 under the 'business-as-usual' (BAU), capital for economy and social well-being (KES), and capital for low-carbon development (KLC) scenarios**

| | Electricity Supply | | | Construction | | | Transport. Svc. | | | Real Estate | | | Residential Svc. | | |
|---|---|---|---|---|---|---|---|---|---|---|---|---|---|---|---|
| | BAU | KES | KLC | BAU | KES | KLC | BAU | KES | KLC | BAU | KES | KLC | BAU | KES | KLC |
| PBE | 4443 | 4862 | 4038 | 78 | 79 | 77 | 717 | 725 | 901 | 15 | 15 | 15 | 11 | 11 | 10 |
| $PBE^K$ | 3105 | 3499 | 2808 | 32 | 32 | 30 | 763 | 769 | 867 | 595 | 598 | 546 | 398 | 401 | 359 |
| $F^K$ from 1995–2017 | 42(1%) | 43(1%) | 40(1%) | 13(42%) | 13(41%) | 13(45%) | 97(13%) | 97(13%) | 97(11%) | 227(38%) | 227(38%) | 227(42%) | 103(26%) | 103(26%) | 103(29%) |
| $F^K$ from 2018–2030 | 110(4%) | 114(3%) | 96(3%) | 17(51%) | 17(51%) | 14(48%) | 188(25%) | 189(25%) | 162(19%) | 358(60%) | 361(60%) | 310(57%) | 286(72%) | 289(72%) | 247(69%) |
| CBE of final consumption | 2381 | 2784 | 2158 | 11 | 11 | 9 | 514 | 519 | 606 | 22 | 21 | 20 | 43 | 43 | 39 |
| CBE | 2381 | 2784 | 2158 | 2576 | 2604 | 2112 | 544 | 549 | 639 | 31 | 31 | 29 | 45 | 44 | 40 |
| $CBE^K$ | 2505 | 2922 | 2274 | 421 | 425 | 379 | 725 | 731 | 801 | 492 | 493 | 451 | 266 | 263 | 239 |
| $F^K$ from 1995–2017 | 35(1%) | 38(1%) | 35(2%) | 122(29%) | 122(29%) | 119(31%) | 69(10%) | 69(9%) | 70(9%) | 181(37%) | 181(37%) | 181(40%) | 60(23%) | 59(22%) | 60(25%) |
| $F^K$ from 2018–2030 | 89(4%) | 100(3%) | 82(4%) | 289(69%) | 292(69%) | 251(66%) | 141(19%) | 142(19%) | 125(16%) | 290(59%) | 291(59%) | 250(55%) | 163(61%) | 161(61%) | 141(59%) |

$F^K$ represents supply chain-wide $CO_2$ emissions embodied in annual capital depreciation. We select four of the key sectors of capital development in China (see Fig. 1). The percentages in brackets represent the share of associated $CO_2$ emissions of sectoral total emissions.

approach to quantify and allocate supply chain-wide capital inputs and associated $CO_2$ emissions among sectors, across regions, and over time. 'Historically committed' $CO_2$ emissions are defined in this study. 'Historically committed' $CO_2$ emissions provide a scheme to assign emission responsibilities of capital activities over time. This over-time accounting scheme allocates capital-associated emission responsibilities to capital users instead of capital producers as conventionally done. Furthermore, this over-time accounting scheme allocates emissions of capital formation from the year of formation over capital's entire lifetime, which relieves emission responsibility for the year of formation yet indeed entails a delay in carbon mitigation. Learning from economic loans or mortgages (used to purchase products in a certain year but paid back over time), the idea of mortgages of emission burdens—emitted during construction but complemented with emission-neutrality measures throughout the payback period to offset emission burdens of the formation year—can reduce the impacts of the delay. This mortgage idea of emission burdens is hence in line with the concept of carbon neutrality. In addition, mortgages of emission burdens are more relevant to investment plans of capital-intensive but climate-friendly projects such as infrastructure for any renewables, if current huge emission burdens are an important concern to launch such projects. There is also a need to avoid climate-unfriendly projects to adopt this mindset and could use it for greenwashing purposes when applying the mortgage idea of emission burdens. Our findings suggest to decision-makers to pay attention to this inertia of capital assets in terms of historic and future 'committed' environmental pressures when making capital investment plans and designing capital-related policies.

Capital assets influence the attainment of all of the Sustainable Development Goals[2]. However, there are only few studies that systematically project future capital development and even less that analyze supply chain-wide downstream environmental pressures. This study fills this important research gap through developing the BAU, KES, and KLC scenarios to compare China's future capital development pathways and associated $CO_2$ emissions. China has promised to peak its $CO_2$ emissions by 2030. To achieve this target, it is projected that China's energy and $CO_2$ intensity levels need to decline by 43 and 45%, respectively[20]. This indicates that a substantial amount of investments in low carbon technologies are expected in the near future, and associated $CO_2$ emissions are mostly in the capital investment rather than the use phase. The KLC scenario presents an alternative pathway for China low-carbon development via efficiency improvement and energy transition. Results show that low-carbon technology investments designed in the KLC scenario would be cost-efficient at the national scale——an extra 7% of low-carbon technology investments would gain a 9% decline in national $CO_2$ emissions compared with the BAU scenario——and in most provinces. In addition, we find that 'historically committed' $CO_2$ emissions are mostly attributable to the production and consumption of capital-intensive service sectors (Table 1), which are usually not regarded as main $CO_2$ emitters. Failing to include this 'historically committed' part of $CO_2$ emissions in $CO_2$ emission accounting especially of service sectors hence strongly underestimates their contributions. Recent news indicates that in light of the COVID-19-induced slump in the world economy[21] and the Russia-Ukraine war[22], China has further stimulated investment, mostly in the service sectors, energy, and food products. The endogenization of capital is an additionally necessary step to ensure that policy makers realize the synergies and trade-offs between capital-intensive economic development and associated environmental burdens, to avoid any 'lock-in' effects[23] on carbon emissions or resource requirements and pursue a cost-efficient pathway of economic recovery like the one demonstrated in the KLC scenario instead of the KES scenario.

China has launched its first national emissions-trading scheme on 16 July 2021[24] with a focus on direct emissions. The direct emissions of electricity generation sector in the future, regarded as future

'committed' emissions[16,17], have been found as its main $CO_2$ emissions (Table 1) compared to the 'historically committed' emissions. However, the choice of conventional emission accounting or the accounting scheme over time considering capital-associated $CO_2$ emissions influences the accounted emissions considerably (Fig. 2 and Table 1), which will finally determine the emissions allowances of each plant in the emissions-trading market. For instance, using the $CO_2$ accounting scheme proposed in this study, an average 30% reduction from conventional PBE of energy sector in 2030 was found under different capital scenarios (Table 1). The changes in emission allowances cannot be specified based on the current study but could result from a similar dedicated scenario study. Nevertheless, we provide suggestions for policy makers to consider capital-associated emissions in China's emissions-trading market, particularly for energy plants: (1) constructing a systematic database[25] that covers the lifetime of each device (start, retired or ceased operating date), fuel types, and generating capacities, which determine the emissions during the operating phase; (2) developing a standard accounting method to quantify capital inputs at high resolution of assets especially for power plant structures, generating devices, and transmission lines[26], which reflect indirect emissions related to capital inputs, that is, 'historically committed' emissions; and (3) formulating a fair price mechanism for both historic and current emissions for companies to trade their emission allowances, which help take into consideration both emissions that are related to energy plants and their roles in the emissions-trading market.

## Methods

### Developing capital database and constructing capital depreciation flows

Official capital investment data from the National Bureau of Statistics of China (NBSC) are recorded by two main annual series, 'total investment in fixed assets (TIFA)' and 'newly increased fixed assets (NIFA)'. TIFA refers to the 'workload' of activities in construction and purchases of fixed assets in monetary terms, which may not produce results that meet standards for fixed assets in the current period or may take many years to become qualified for fixed assets[27]. NIFA refers to the value of investment projects completed and put into production or meeting the standards for fixed assets in the current year, hence reflecting the fixed assets formed in the current period as a result of those effective investment projects taking place in the current and previous periods. Given that the concept of 'capital investment' used in the perpetual inventory method (PIM), a standard geometric method that is adopted in this study to calculate capital consumption time series, are those effective capital assets that have been completed and put into production, this study relies on NIFA to construct the provincial capital investment time series. More details about the differences between TIFA and NIFA and the problem of directly using TIFA in PIM are discussed in Supplementary Information 6.

Although NIFA (denoted as $N$, in Yuan per year) is more reasonable than TIFA to be used as capital investment (denoted as $I$, in Yuan per year) in PIM, an upward adjustment has to be made to transfer $N$ to $I$. This upward adjustment is to include the projects less than half million yuan by non-state firms that are not reported in official investment statistics plus the value of likely underreported[28]. The standard $I$ by sector $s$ of province $m$ in year $t$ could be estimated as:

$$I_{m,t,s} = \frac{N_{m,t,s}}{1 - \lambda_{m,t,s}}, (\lambda < 1) \qquad (1)$$

where $\lambda$ is to adjust $N$ by the effects of missing and/or underreported investment. There is little information available on $\lambda$ especially those at provincial level. We apply the national $\lambda_{t,s}$ from Wu[29] to adjust $N_{m,t,s}$, and further scale $N_{m,t,s}$ into the national capital investment by sector $s$ in year $t$ from WORLDKLEMS. We also specify 37 sectors (Table S3) in

our provincial capital investment dataset, which are consistent with the sectoral classification in WORLDKLEMS.

There are limited investment data by asset type especially at industrial level. In the official investment statistics, under the sub-categories of TIFA 'capital construction' and 'technical update and transformation', there are data for 'equipment' and 'structures'. The 'structures' indicator also distinguishes 'housing' or 'non-productive' constructions. We rely on TIFA by these categories (although they are not directly relevant with NIFA), and industrial investment statistics in annual statistics bulletins[30] about industry and transportation economy, commune and brigade factories, and township and village enterprises to disaggregate the capital investment. According to Wu[29], this study also disaggregates four categories of industry-specific fixed assets, namely, 'equipment', 'residential structures', 'non-residential structures' and 'others'. We re-allocate 'others' into 'equipment' and 'non-residential structures' by a ratio of 3:7 according to Wu[29]. Without category-specific data on investments in non-industrial sectors (i.e., agriculture, construction, and all services), we assume that the non-industrial sector-specific $I$ is equal to the official NIFA of that sector. We use the share of productive structures given by the economic-wide TIFA to decompose the total investment into non-residential structures and equipment.

The procedures to trace and allocate the contribution of year $t$'s capital investment to year $n$'s inter-industrial production networks follow the global capital endogenized MRIO model[14]. In this study, we develop a Chinese version of the capital endogenized MRIO model (details about the procedures can be found in Supplementary Information 9). This Chinese version relies on the inter-provincial MRIO tables of China and focuses on the impacts of capital development in China on the emission responsibilities of provinces across China. The international import and export linked to other countries are aggregated in the MRIO tables of China. The key step to obtain the supply chain-wide capital consumption matrix $\mathbf{D}_{t,n}^K$ ($t \le n$, in Yuan) within China is re-creating the concordance tables that are used to convert capital assets and capital consumption sectors (37 sectors, Table S3) into the sectoral classifications of MRIO tables (42 sectors, Table S6). The final capital consumption matrix $\mathbf{D}_{t,n}^K$ within China has capital producing and capital consuming sectors along rows and columns, respectively; and each element records the quantity of assets that were invested in year $t$ and consumed (i.e., depreciated) in year $n$. It should be noted that the capital flows showed in Fig. 1a are in the unit of 2017 US dollars. We rely on the price indices like currency convert rates, consumption price index of US dollars obtained from The World Bank to convert Yuan to 2017 US dollars.

## Constructing China's inter-provincial MRIO table series (1995−2017)
We rely on the current best available MRIO tables in 2007[31], 2010[32], 2012[33], 2015 and 2017 from CEADs[34], 1995−2006 from Wang[35] as the benchmarks to construct the inter-provincial MRIO table time series. Before that, we first adjust the final demand, exports, imports and value-added data in the benchmarking MRIO tables, according to the available statistical data from the NBSC. This is because we found that some benchmarking MRIO tables have big data gaps based on available statistical data, especially for early years. We rebalance these benchmarking MRIO tables using the GRAS method[36], and use the benchmarking MRIO tables to estimate the MRIO table in the missing years. Details about estimating final demand, exports, total outputs, and using the GRAS method to balance the MRIO tables in the target years can be found in Supplementary Information 8.

## Re-allocating capital-associated CO₂ emissions ($F^K$)
The supply chain-wide CO₂ ($\mathbf{F}_{t,n}^K$, in tonnes) that emitted in year $t$ when the capital inputs allocated to year $n$'s production activities (i.e., $\mathbf{D}_{t,n}^K$) can be estimated by conventional IO modelling, $\mathbf{F}_{t,n}^K = \hat{\mathbf{S}}_t \mathbf{L}_t \mathbf{D}_{t,n}^K$, where

$\mathbf{S}_t$ (in tonnes per Yuan) is a row vector of direct CO₂ emission intensities of economic activities, collected from CEADs[37–39]; $\mathbf{L}_t$ (Yuan per Yuan) is the Leontief inverse matrix[40]. When allocating $\mathbf{F}_{t,n}^K$ to the actual capital using sectors in year $n$, we can obtain production-based emissions of capital depreciation in year $t$ for year $n$'s production. The allocation of $\mathbf{F}_{t,n}^K$ to capital using sectors as production-based emissions follows the conventional logic of production-based emission assignment (details see Supplementary Information 15). When allocating $\mathbf{F}_{t,n}^K$ to final consumption ($\mathbf{Y}_n^{FC}$, including final expenditure of rural population, urban population, and government, in Yuan), gross fixed capital formation ($\mathbf{Y}_n^{GFCF}$, in Yuan), and international exports ($\mathbf{Exp}_n$, in Yuan) in year $n$, we can obtain capital-associated consumption-based emissions in year $t$ for different economic activities in year $n$ (Eqs. 2–4).

$$\mathbf{S}_{t,n}^K \mathbf{L}_n \mathbf{Y}_n^{FC} \tag{2}$$

$$\mathbf{S}_{t,n}^K \mathbf{L}_n \mathbf{Y}_n^{GFCF} \tag{3}$$

$$\mathbf{S}_{t,n}^K \mathbf{L}_n \mathbf{Exp}_n \tag{4}$$

where $\mathbf{S}_{t,n}^K$ (in tonnes per Yuan) describes the one-unit CO₂ emissions of province-sector pairs in year $t$ that consumed $\mathbf{D}_{t,n}^K$ in year $n$, calculated by $\mathbf{S}_{t,n}^K = \varphi \mathbf{F}_{t,n}^K \hat{\mathbf{x}}_n^{-1}$, in which $\mathbf{x}_n$ (in Yuan) is a column vector of total economic outputs of year $n$, and $\varphi$ is a summation vector of ones.

## Re-assessing CO₂ emissions of provinces
Different from conventional emission accounting of capital activities (represented by consumption-based CO₂ emissions of GFCF in one-year base, $CBE_t^{GFCF} = \mathbf{S}_t \mathbf{L}_t \mathbf{Y}_t^{GFCF}$, in tonnes), we re-allocate $CBE_t^{GFCF}$ to the actual capital using sectors or further to the final demand throughout the assets' lifespans according to annual $\mathbf{F}_{t,t}^K$, $\mathbf{F}_{t,t+1}^K$, $\mathbf{F}_{t,t+1}^K$, … This re-allocation of $F^K$ hence changes annual CO₂ emission accounting of provinces from both production-based and consumption-based accounting (i.e., consumption-based CO₂ emissions of final demand, $\mathbf{S}_n \mathbf{L}_n (\mathbf{Y}_n^{FC} + \mathbf{Y}_n^{GFCF} + \mathbf{Exp}_n)$). Two steps are taken to re-assess provincial CO₂ emissions. One is omitting the conventional PBE and CBE that are related to GFCF of a province. The other is adding back $F^K$ re-allocated to capital using sectors generating PBE after $F^K$ re-allocation (PBE$^K$), or adding back $F^K$ re-allocated to final demand generating CBE after $F^K$ re-allocation (CBE$^K$). Units of PBE, CBE, PBE$^K$, and CBE$^K$ are tonnes.

## Developing scenarios of capital investment until 2030
We project China's capital investment pathways by two scenarios into 2030, and a 'business-as-usual' (BAU) scenario as the baseline scenario. The two capital investment pathways are developed on the principle of improving economic growth and social well-being (KES), and the principle of low-carbon development (KLC), respectively. All three scenarios are in constant prices of year 2017 and developed based on the economic activities and CO₂ emissions in 2017 as the base year. We assume that the national average capital intensity of one-unit GDP is the same for each scenario. All the scenarios will be implemented by manipulating the MRIO tables of the year 2017 to each projecting year. Details of each scenario are summarized in Table S7.

The BAU scenario, referred to De Koning et al.[41], is developed by continuing historical trends of population growth, efficiency improvements, and productivity growth until 2030 (summarized in Table S8). The trends in general efficiency improvement (influenced by current economic and climate policies) into 2030 are determined by actual trends in the last decade, looking in detail at sector- and province-specific development (recorded in $\mathbf{A}$). If we assume total outputs in a projected future year would not change, efficiency

improvements reduce intermediate inputs (including domestic inputs and imported inputs) for economic activities and further lead to substantial economic growth. We then make up the difference to meet overall GDP growth (recorded in **vd**) based on the autonomous economic growth accomplished by efficiency change. GDP growth rates are set as 6.5% per year before 2020, and 5% per year after 2020[42,43]. Final consumption of rural and urban population (recorded in $\mathbf{Y}^{FC}$) is estimated based on projected population, urbanization rates, and per-capita expenditures. Rural and urban population in each province until 2030 are estimated using total provincial population and national urbanization rate, obtained from Chen et al.[44]. Per-capita final expenditure of rural and urban population until 2030 are estimated by the same method used in previous studies[45,46]. Final consumption of government is estimated according to the total changes in the final consumption of rural and urban population. Capital investment of sectors is estimated according to required future capital stock of each sector. We first predict the capital stock intensity of value-added of each sector in 2030, based on its capital stock intensity in the base year, elasticity parameter, and changes in capital price[47]. Total capital stock of each sector in 2030 can be calculated by multiplying the capital stock intensity with its total value-added. Annual average capital investment until 2030 can be calculated by the PIM, given that the capital stock in year $T$ equals to $\sum_{t}^{T} I(1-\delta)^{T-t+1}$, where $\delta$ is the depreciation rate of each asset. After that, we distribute the capital investment of each investing sector to capital producing sectors, based on the capital production structure in 2017, to obtain $\mathbf{Y}^{GFCF}$ in target year. International export is assumed to proportionally increase according to growth of GDP. Total outputs, intermediate inputs, and international imports for intermediate inputs in the target year can then be calculated by the basic equations of IO modelling. We balance total inputs and outputs through GRAS method (Supplementary Information 8). Furthermore, we adjust the balanced MRIO tables according to the changes in energy mix (Figure S6). The total energy supply and use are consistent with the projections in IEA[48]. Changes in energy supply and use per source (i.e., coal, oil, natural gas, nuclear and renewable energy) lead to proportional changes in the transactions with associated sectors (e.g., coal mining). We also adjust the intermediate inputs from different energy sources to electricity generation sectors according to the changes in their shares in total power generation. The adjusted MRIO tables are balanced again using GRAS method. Direct $CO_2$ emissions from sectors are changed as well (recorded in **F**). It is assumed that the changed intermediate inputs in sectors bring changes in emissions accordingly.

Under the KES scenario, China is increasingly focusing on the role of capital assets, especially infrastructure, to improve economic growth and social well-being[49]. We rely on the associated outlook of infrastructure development in China (summarized in Table S8) from GIH[1], and integrate future infrastructure investment data into the MRIO model for $CO_2$ emission accounting. Seven infrastructure categories are covered in this scenario, i.e., roads, railways, airports, seaports, electricity generation and supply, water generation and supply, and telecommunications. The KES scenario is developed on top of the BAU scenario. We first determine the sectors that invest in the associated infrastructure. That is, we assume roads, railways, airports, and seaports are mainly based on investments by the sector '*Transportation, storage and post services*', electricity/water generation and supply are invested by the sector '*Production and supply of electricity, heat, gas, and water*', and telecommunications are invested by the sector '*Information transfer, software and information technology services*'. According to statistical data recorded in NBSC[50], investments in roads, railways, airports, and sea ports annually account for approximately 94% of total investment from '*Transportation, storage and post services*', investment in electricity/ water generation and supply accounts for approximately 97% of total investment from '*Production and supply of electricity, heat, gas, and water*', and investment in telecommunications annually accounts for

approximately 92% of total investment from '*Information transfer, software and information technology services*'. For each infrastructure category, we disaggregate its investment into three assets (see section Disaggregating capital investment by asset type). Asset-specific investment data will be allocated to capital producing sectors according to sectoral shares in GFCF, obtaining a GFCF matrix that represents GFCF of capital producing sectors to build the infrastructure. Furthermore, we scale the GFCF matrix of infrastructure according to annual shares of infrastructure investment in total investment from associated investing sectors. Based on the scaled GFCF matrix, we further adjust GFCF of infrastructure producing sectors under the BAU scenario (if there is any investment default) to get $\mathbf{Y}^{GFCF}$ under the KES scenario. Final consumption from seven infrastructure-related sectors will change proportionally according to their investment. We assume more investment in specific infrastructure will lead to more consumption (evidence to support this assumption can be found in Fig. S7). Value-added in this scenario would change due to changes in final demand, compared with the BAU scenario. Intermediate inputs would also change to meet economic production of final demand and exports. The adjusted MRIO tables are balanced again using GRAS method. Direct $CO_2$ emissions from sectors are changed as well. It is assumed that the changed intermediate inputs in sectors compared with those in the BAU scenario changes emissions accordingly.

The KLC scenario is designed to focus on China's future capital investment in low-carbon technologies by the electricity generation sector, and end-use sectors such as transportation services. Detailed data (Table S8) for energy supply and energy use by energy sources (i.e., coal, oil, natural gas, nuclear and renewable energy), capital investment requirements on low-carbon technologies (e.g., carbon capture and storage, or electric vehicles) by different using sectors (e.g., industry sectors, or transportation services), $CO_2$ emissions by economic sectors are collected from the World Energy Outlook 2017[48]. The procedures to construct $\mathbf{Y}^{GFCF}$ matrix under the KLC scenario according to the capital investment in low-carbon technologies and further adjustments on $\mathbf{Y}^{FC}$ matrix are described in the development of the KES scenario. Changes in energy mix in the MRIO tables follows Fig. S6, which has been described in the development of the BAU scenario. International export would decline whereas international import would increase[51], since the objective of this capital investment pathway is to reduce China's territorial $CO_2$ emissions. The adjusted MRIO tables are balanced again using GRAS method. Lastly, we further adjust the direct $CO_2$ emissions from sectors accordingly, based on emissions data from the World Energy Outlook. Only relative changes in key parameters such as $CO_2$ intensity per unit of GDP are used in developing the KLC scenario.

## Limitation and future work

This study has several limitations. Our model relies on data from multiple sources with different levels of uncertainty, such that the calculated results need to be interpreted with caution. For instance, it is more reasonable to identify temporal changes of capital-associated $CO_2$ emissions, relative importance of final consumption categories, or differences in results produced by different models. Second, evaluating the use of physical capital assets in production by capital consumption or capital services is highly debated. Data on capital consumption are more readily available than that of capital services. In addition, capital services rely on the prices of capital assets which have higher uncertainty among provinces and for different years. Capital consumption data are frequently calculated using the PIM, which has been widely accepted by national and international statistical agencies and researchers. Thus, we also conduct this analysis by relying on capital consumption to represent the use of physical capital assets. The limited number of categories of capital assets also raise uncertainty in asset-specific depreciation and emission properties. Miller et al.[52] showed that using detailed capital assets versus aggregated KLEMS

assets leads to very different capital-input coefficients. This limitation is in line with a main limitation of IO modelling to aggregate products with different environmental properties into homogeneous sectors. This limitation hence states the importance to build such a capital database with a high resolution of types of capital asset and investing sectors. Another way to construct capital inputs in economic production is based on good life cycle inventory (LCI) databases which record inputs of key capital assets such as infrastructure, machinery, ICT, and etc[53]. One may argue that only lifespan-average capital inputs are recorded in most life cycle assessment. In this case, applying a proper approach in quantifying capital depreciation time series (e.g., the PIM) is a requirement in developing LCI databases. Third, when we develop capital investment scenarios, final consumption of capital investing and using sectors are further adjusted by their contributions in capital investment. A linear correlation is applied between sectors' capital investment and final consumption, based on the correlations observed from electricity and water supply sector, and transportation services sector (Fig. S7). Feedback of investments and side effects on consumption in other sectors are also neglected. This drawback arises from the static feature of the MRIO model. Integrated with a dynamic economic model, to some extent, can reduce the uncertainty from dynamic changes in capital investment and final consumption among all the economic sectors. Lastly, shifting the fossil-based economy to a renewable energy-based economy and achieving carbon peak and neutrality will request large capital investments in low-carbon equipment and infrastructure. Other alternative pathways of capital development are also interesting for future exploration, yet need other methods and approaches to model the entire economic and energy structures for future capital development narratives.

## Data availability

The capital investment (*I*) data generated in this study have been deposited in the public repository Figshare with the identifier https://doi.org/10.6084/m9.figshare.20407572[54].

## Code availability

The source codes for data processing and capital-endogenization model are available at https://github.com/yequanliang1993/capital-endogenized-input-output-model.

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

## Acknowledgements

The time and comments from A. E. Steenge are gratefully acknowledged. Q.Y. is grateful for the scholarship he received from the China Scholarship Council (CSC), No. 201806710143. The time and comments from the editor and three referees are gratefully acknowledged.

## Author contributions

Q.Y., M.S.K, Y.S and K.H. designed the research. Q.Y. developed the model; Q.Y., M.S.K., Y.S., and K.H. analyzed the data. Q.Y., M.S.K., J.F.S, and M.B. discussed the results and designed the figures. Q.Y., M.S.K, Y.S., J.F.S, M.B., and K.H. wrote and revised the main manuscript.

## Competing interests

The authors declare no competing interests.
