## [Peer Review File · Nature Communications]

Allocating capital-associated CO₂ emissions along the full lifespans of capital investments helps diffuse emission responsibilityREVIEWER COMMENTS

Reviewer #1 (Remarks to the Author):

This study is trying to allocate carbon emissions of capital assets over time. I agree that built capital is important in carbon emissions as it links the emissions in the past, present and future. However, this study fails to answer the questions of how this reallocation is linked to climate effect, carbon mitigation policies, and climate actions.

1, the relocation over the lifetime is different from the reallocation of emissions between producers and consumers. The reallocation from producers to consumers because it is clear consumers can contribute to carbon reduction by the change in demand. But how the reallocation from the formation year to the future can contribute to carbon emissions, even you argued "allocating CO2 emissions of capital assets over time relieves emission responsibility for the year of formation"? Who will take the reallocated responsibility?

2, the findings that "the existing capital in 2017 will still contribute approximately 10% of China's CO2 emissions in 2030" is misleading. China's carbon emissions in 2030 is the allocation in your study, not the physical carbon emissions. The emissions also can not be reduced in 2030 as it was released in 2030. Furthermore, if the emissions are relocated to future years, how we can achieve carbon neutrality.

3, Figure 2. The aggregated yearly emissions are meaningless. It is important to show how the capital formed in this year affects the emissions in another year.

4, The results and discussions are too descriptive, there are no implications on how this allocation can affect climate change as the emissions are physically released in the formation year. It is also not clear how this reallocation is linked to carbon mitigation policies and climate actions.

Overall, this paper is too technical and descriptive, it is more suitable for a disciplinary journal.

Reviewer #2 (Remarks to the Author):

Overall Comments:

The authors have performed a detailed and thorough analysis, building on prior work to demonstrate the importance of endogenizing capital assets into MRIO models with a case study of China, and thus offer a valuable contribution. In the comments below, I have a few questions about the methodological approach. I am in support of the consumption-based accounting approach, but am not yet convinced by the modified pseudo- production-based accounting approach. My other feedback is mainly to ensure that these complex concepts are explained thoroughly and clearly for the reader throughout, reducing any ambiguity in terminology and phrasing.

L25:

While capital assets contribute to CO2 emissions over their lifetime, that is due to either the use phase impacts or embodied impacts of maintenance and repair. This is different from the concepts being discussed in the paper, which instead argue that embodied CO2 emissions ought to be allocated across the lifetime. This concept arises again in Figure 1b, with the term "Emissions of Capital Use". Consider incorporating the term "embodied" throughout.

L48-50:

Please rephrase or reorganize this sentence to more clearly distinguish capital assets from non-capital assets; the producers of any good are usually different from their users, so this is not a clear distinction.

The related Lines 101-104 are more clear, but there the phrase "for the production of final demand" is inaccurate, rather "for production to satisfy final demand". This appears again in 112-113.

L53-57:

Please consider rephrasing to reduce confusion. Many assets are partially depreciated, but the reader might infer that instead 1/3 of assets are fully depreciated and 2/3 are new. Also, while depreciation may be due to the physical effects suggested, there are other non-physical reasons (technological, cultural) for an asset becoming obsolete, and financial depreciation estimates may/not align with physical depreciation estimates.

89-91: While the Methods does describe the scenarios in depth, since they are frequently referred to in prior text, it would help the reader to have a somewhat longer description here.

L123: Stating "capital use and depreciated capital" is redundant, since use of capital is equivalent to capital depreciation.

L182-184: This is an interesting paragraph. However at the start, since conventional and re-allocated PBE and CBE accounts are both divided by the same population to get per-capita results, a relative decline in the absolute results would necessarily imply a relative decline in the per-capita results.

L240-242: Consider incorporating the concept of "carbon lock-in" (DOI 10.1146/annurev-environ-110615-085934) with the concept of future 'committed' CO2 emissions.

L254-257: The result of 83% for real estate is consistent with the 77%-86% for residential housing found by Berrill et al. (2020) with a capital-endogenized model of the US (<https://doi.org/10.1111/jiec.12953>). It could be helpful to clarify for the reader that the remaining 17% likely reflects embodied impacts of maintenance and repair along the lifespan (considered non-capital goods), rather than a possible interpretation that the residential use phase impacts are only estimated to be 17% of the life cycle carbon footprint of a residential building. The residential use phase impacts are typically accounted for separately in input-output via consumption of electricity etc.

Figures:

In Figure 1:

- I like that you cleverly distinguish the time periods in the "Consumption Perspective" nodes.
- I would suggest combining the redundant "Capital Investment" and "Capital Use" columns into "Capital Investment & Use".
- Also, it would be easier for the reader to trace the flows if they were colored the same as the source or target nodes, rather than monochrome.

In Figure 2:

- The caption helps explain the series, but it would be easier to interpret if there was a symbol to distinguish the PBE and CBE without re-allocated capital, such as PBE* and CBE*.
- In 2b, Please clarify what the reference values are for the percent changes, as it is not readily apparent from the chart and does not read left-to-right.

Methods:

L156: While I agree that endogenizing capital is indeed consistent with consumption-based accounting, I am concerned that re-allocating supply chain-wide embodied emissions of capital use is not consistent with production-based accounting principles, as mentioned briefly in L394-395. Peters & Hertwich (2008) define: "Production = GHG emissions from resident institutional units" (DOI 10.1007/s10584-007-9280-1).

From my perspective, conventional production-based accounts offer an important and complementary perspective to capital-endogenized consumption-based accounts. The former look at the emissions where and when they occur, and the latter look at final demand drivers of those emissions. If the production-based accounts are manipulated as proposed, the perspectives start to blur, and

responsibility becomes less clear. To me, the article needs to provide a stronger argument for its use.

L331 - 342: This is a very interesting discussion of TIFA vs NIFA. Can you please describe how the embodied impact of investments that do not become fully functional (those that 'are completely wasted' as discussed in SI section F) are reflected in your accounting approach? Or are they insignificant?

L354 - 367: The approach to disaggregate capital investment by asset type is often a challenge when endogenizing capital assets, given the lack of detailed data. I appreciate efforts made here to make the best of the data, and calls for more detailed data in Lines 323-328 (which could be expanded). However, please describe this shortcoming in the Limitations section. For instance, Miller et al. (2019) show that using detailed capital assets versus aggregated KLEMS assets leads to a very different capital IO table (DOI 10.1111/jiec.12931).

L370: I am unclear how other countries are considered in this model. This line 370 states: "We adapted the global capital endogenized MRIO model into a Chinese inter-provincial case", while L84-87 states: "We first develop an inter-provincial capital-endogenized multi-regional input-output (MRIO)". The SI tables focus on provinces and does not list the foreign country resolution (which is important, since lower country resolution could impact accuracy).

Relatedly, there is minimal discussion throughout the paper on China's role as a chief global exporter, and the influence that has on differences between CBE with and without capital endogenized. Assuming this is a global model, it would be useful to explain that the amount of emissions in this gap would be re-allocated for production to satisfy other countries' final demand.

L388-403: I noticed "Fk allocated to GFCF" in Figure 2a. Please carefully consider if there is double-counting embedded in associated equation 3 in L402. If the GFCF is fully endogenized (and therefore removed from Y), then this approach would seem to result in the embodied impacts of that capital being accounted for twice over time.

SI:

Figure S1: To me it would be more appropriate to label "GFCF in final demand (%)" rather than "in Value-Added", since GFCF is a component of final demand, while CFC is a component of Value-Added.

Figure S4: Please provide a legend for the colors.

Typos (and their corrections):

L122: precious (previous)

L396: constant (consistent)

SI Section A: "The lack (?) of Ye et al.3 is the lack of understanding how the existing capital assets will sever (serve) future"

Reviewer #3:

This article discusses the reallocation of impacts from capital investments to the capital-demanding sectors, regions, or time-periods.

I am impressed by the amount of data and the way the authors have processed all this information into a concise manuscript.

At the same time, I am wondering why they are undertaking this exercise. Figure 1 illustrates the difference between a production and consumption perspective. I would claim that in a production perspective, the producing sector is responsible, while in a consumption perspective the consuming sector is responsible. Therefore, the impacts from capital formation are to be allocated to the capital-producing industry in a production perspective, and to the final consumer in a consumption perspective. So there would be no need to reallocate the impacts from capital investments to capital-demanding sectors. I understand the article as an attempt to create a sort of in-between thing: reallocating capital to producing industries. I may be wrong in this perception, but in any case, the article now reads in that way. Take the sentence "Our study reveals that conventional estimations of supply chain-wide CO2 emissions of 'capital investment' are, to some extent, misleading the allocation of capital-associated emission responsibilities to capital producers instead of capital users." (lines 117-119.) Words like "misleading" suggest that the an incorrect answer is given to a certain question, but that obviously depends on what the question is. If we want to know "who is emitting how much", the production perspective is clearly right, without any reallocation. If we want to know "which final demand is responsible for how much", we need to reallocate production and capital to the final use. Also lines 133-135 ("When re-allocating this part of capital-associated CO2 emissions to the actual capital consumers or further to final goods and services throughout the full lifespan of capital, it would substantially alter CO2 emission accounting at both regional and sectoral levels") testify the confusion and lack of clarity about the purpose of the whole exercise.

My main concern is therefore that it is unclear which question(s) the authors have in mind.

Another point is that I miss the connection with life cycle assessment (LCA). In contrast to MRIO, an LCA is not based on an accounting year, and there are therefore no temporal reallocations needed, even while LCA takes a consumption perspective. If a capital good is needed that depreciated in 20 years, and every year the factory produces 50,000 products, then the per-product input of capital is 1 millionth. Good LCA databases include such information. See also DOI 10.1111/jiec.13271. I think the temporal reallocation is much more critical than the sectoral or regional reallocation. It might be worthwhile to focus more on this time aspect, and on the way it has been "solved" in LCA.

In that sense, I also point to the confusion that is introduced in lines 272-278. This text starts by mentioning two topics, the second of which is the "temporal displacement of environmental responsibilities along capital's lifespan". Next the text says that "This study explores the second issue, and demonstrates a new approach to quantify and allocate supply chain-wide capital inputs and associated CO2 emissions among sectors, across regions, and over time." So, is time alone the second topic, or is the triple {sector, region, time} the second topic?

Some smaller points are the following:

- * Line 339 mentions "the perpetual inventory method (PIM)", which is not further explained. The SI provides a reference to PIM, but some more information would be appreciated.
- * It would be helpful if symbols (like those in line 348 and 389) would not only be described, but if also units could be provided. For instance, is I in line 348 in yuan? In yuan/yr? In another unit? This applies to all symbols, also in the SI.
- * Please be a bit more precise in the graphs. Figure 2a has a vertical "Gt" and 2b has "CO2 emission per capita (tonnes)". I guess 2a is also CO2 emissions? Moreover, the lines in 2a look pretty smooth, while I think you had only 20 points (1995, 1996, 1997, etc, not anything in between). Similar remarks hold for Figure 3, as well as for the SI.
- * Table 1 is also without units.

Reviewer #1 (Remarks to the Author):

This study is trying to allocate carbon emissions of capital assets over time. I agree that built capital is important in carbon emissions as it links the emissions in the past, present and future. However, this study fails to answer the questions of how this reallocation is linked to climate effect, carbon mitigation policies, and climate actions.

The authors thank the reviewer for sharing his/her valuable questions and comments. We also greatly appreciate the time and efforts the reviewer generously spent reviewing this manuscript. We felt very sorry that the reviewer didn't give more optimistic assessments on our study. After this round of revision, we believe we have understood and addressed the reviewer's comments and major concern (*how the temporal allocation results related to climate effect, carbon mitigation policies, and climate actions*). We hope the revisions made in this manuscript can alleviate the reviewer's concerns.

We have provided responses (in indented and in black) and revised texts as they appear in the revised manuscript and supplementary information (in *indented and in blue italics*) below each comment the reviewers made (in **black italics and bold**). The line numbers in this response letter correspond to those in the "tracked-changes" version of the revised manuscript or supplementary information.

1, the relocation over the lifetime is different from the reallocation of emissions between producers and consumers. The reallocation from producers to consumers because it is clear consumers can contribute to carbon reduction by the change in demand. But how the reallocation from the formation year to the future can contribute to carbon emissions, even you argued "allocating CO2 emissions of capital assets over time relieves emission responsibility for the year of formation"? Who will take the reallocated responsibility?

Thank the reviewer for asking these two important questions. We agree with the reviewer that the reallocation of emissions over the lifetime differs from the reallocation between producers and consumers for a one-year accounting period. But the reallocation over the lifetime is still in line with conventional consumption-based accounting. That is, whoever consumes or uses a final product should be responsible for environmental impacts related to the production and consumption of the product. When looking at capital assets, the consumers (e.g., capital using sectors, or final consumers) exist and benefit from capital assets along the entire lifespans of capital assets. As such, it is important to extend the conventional consumption-based accounting by adding a longer time horizon.

We also agree with the reviewer that reallocating emission responsibilities from producers to consumers informs consumers about their responsibility and how they can contribute to carbon reduction by changing their demand. We think there are two important steps to enable consumers to reduce carbon emissions. The first step is applying an accurate allocation method to assign emissions from producers to consumers, which is related to the reviewer's second question (***Who will take the reallocated responsibility?***). Second is what consumers can do to

reduce carbon emissions along the supply chains of their final demand, which is related to the reviewer’s first question (*how the reallocation from the formation year to the future can contribute to carbon emissions?*). We want to answer the reviewer’s second question first.

The answer to the reviewer’s second question is that the final consumers is responsible for the re-allocated responsibilities. Our temporal allocation follows consumption-based accounting but extends the conventional (spatial) allocation with a time dimension. The reason for adding this time dimension is that different from non-capital products, “*capital assets can exist for several years or even decades, and serve economic production throughout their lifespans*” (Line 56–57). A simple example is that passengers still use railway infrastructure built years ago. According to consumption-based accounting, passengers as consumers along the lifespan of railway infrastructure are responsible for the environmental impacts caused by the production of (or ‘embodied’ in) the infrastructure. As shown in Figure 1b (attached below), the supply chain-wide CO₂ emissions embodied in the infrastructure constructed during 1995–2017 are first allocated to capital formation sectors (CBE^{GFCF} , as conventional consumption-based accounting does), and then allocated to final demand (including final consumption, gross fixed capital formation, and export) during 1995–2017 and afterwards. In other words, instead of allocating the responsibility of infrastructure-related emissions to the year of construction, the responsibility is allocated over the years of use. This follows strictly the logic of consumption-based accounting by acknowledging that the final users are responsible for the supply chain effects of consumption but explicitly recognizing that the final users take advantage of infrastructure for the whole duration of the infrastructure.

Figure 1. Monetary capital flows (a) and embodied CO₂ transfers (b) across key sectors of China’s capital development.

To answer the reviewer’s first question, all the measures from the demand side are also effective to reduce CO₂ emissions under our accounting scheme. Our temporal allocation method didn’t change the supply chain-wide emission results of final consumption (shown as the grey areas in Figure 2a, attached below). Instead, our method takes one step further to consider the capital inputs and embodied CO₂ emissions to produce associated final consumption (the purple areas in Figure 2a). We think our approach accounts more comprehensively for emission responsibilities of final-consumption activities, because it captures upstream emissions of both intermediate inputs and capital inputs. The latter is usually neglected in conventional IO-based emissions accounting. In detail, “*we find that capital-associated CO₂ emissions are mostly attributable to the production and consumption of capital-intensive service sectors (e.g., real estate and residential services), which are usually not regarded as main CO₂ emitters*” (Line 339–341). We argue that “*failing to include this capital-associated part of CO₂ emissions in CO₂ emission accounting especially of service sectors hence strongly underestimates their impacts on the climate.*” (Line 341–343). As such, “*effective demand-side measures, such as reducing the consumption of emission-intensive products (e.g., livestock), should be complemented by reducing the consumption of capital-intensive products (e.g., big houses) based on our study*” (Line 343–345). In conclusion, we believe that “*the endogenization of capital is an additionally necessary step to ensure that policy makers realize the synergies and trade-offs between capital-intensive economic development and associated environmental burdens, to avoid any ‘lock-in’ effects on carbon emissions or resource requirements*” (Line 347–351).

Figure 2. Alteration to the production-based emissions (PBE) and consumption-based emissions (CBE) due to capital re-allocation. In (a), the national PBE with (PBE^K represented by the orange dashed line) and without (conventional PBE represented by the red solid line) the re-allocation of capital-associated CO₂ emissions (F^K), as well as the national CBE with (stacked grey, purple and light purple areas) and without (i.e., the conventional CBE represented by the black solid line) the re-allocation of F^K are shown.

Hope the above explanations answer the reviewer's two questions.

2, the findings that “the existing capital in 2017 will still contribute approximately 10% of China’s CO2 emissions in 2030” is misleading. China’s carbon emissions in 2030 is the allocation in your study, not the physical carbon emissions. The emissions also can not be reduced in 2030 as it was released in 2030. Furthermore, if the emissions are relocated to future years, how we can achieve carbon neutrality.

Thank the reviewer for raising these points and asking the question.

Regarding the point “**the findings is misleading**”, truly, it is indeed not the best way to state this finding. The capital does not contribute to actual emissions are but to the emission responsibility. We have rephrased this statement to reflect this important difference. Details are “*Overall, the existing capital in 2017 will still carry approximately 10% responsibilities of China’s CO₂ emissions in 2030*” (Line 35–36).

As for the reviewer’s second point, “**The emissions also cannot be reduced in 2030 as it was [not] released in 2030**”, We agree that the re-allocated capital-associated CO₂ emissions have no influence on actual CO₂ emitted each year. But we “*provide a new scheme to assign emission responsibilities of capital activities over time*” (Line 308–309). The necessity to conduct such a temporal allocation was fully explained in our responses to the reviewer’s comment #1. We suggest that “*effective demand side measures, such as reducing the consumption of emission-intensive products (e.g., livestock), should be complemented by reducing the consumption of capital-intensive products (e.g., big houses) based on our study.*” (Line 343–345).

To answer the reviewer’s last question (**how we can achieve carbon neutrality?**):

1. to the question of what kind of policy suggestions can be provided from the temporal allocation scheme under the background of carbon neutrality, we propose a new scheme to make capital development plans, especially for those capital-intensive but climate-friendly projects such as hydropower dams. We refer to the mortgage of economic loans, which are used to purchase products in a certain year but paying back over time. When current huge emission burdens are the main concern to launch such a capital-intensive but climate-friendly project, the idea of mortgages of emission burdens—emitting currently but complementing with emission-neutrality measures throughout the payback period to offset emission burdens of the formation year—can be applied to the project. In this context, our accounting scheme is in line with the concept of carbon neutrality, and deepens the neutrality processes by considering a broader time scope. The added content in the revised manuscript is as follows: “*Historically committed’ CO₂ emissions provide a new scheme to assign environmental responsibilities of capital activities over time. The new accounting scheme allocates capital-associated environmental responsibilities to capital users instead of capital producers as conventionally done. Furthermore, the new accounting scheme allocates emissions of capital*

formation from the year of formation over capital's entire lifetime, which relieves emission responsibility for the year of formation yet indeed entails a delay in carbon mitigation. But one can argue that the mitigation is to be done when the benefit is accrued rather than upfront at the point of investment. Learning from economic loans or mortgages—used to purchase products in a certain year but paid back over time, our new accounting scheme offers decision makers new insights into the investment plans of capital-intensive but climate-friendly projects such as infrastructure for any renewables. When current huge emission burdens are an important concern to launch such a capital-intensive but climate-friendly project, the idea of mortgages of emission burdens—emitted during construction but complemented with emission-neutrality measures throughout the payback period to offset emission burdens of the formation year—can be applied to the project. In this context, our accounting scheme is in line with the concept of carbon neutrality. Our findings suggest to decision-makers considering this inertia of capital assets in terms of historic and future 'committed' environmental pressures when making capital investment plans and designing capital-related policies.” (Line 307–327).

2. with regards to what alternative pathways of capital investment for achieving carbon neutrality, it is not the focus of our scenarios. We cannot tell an alternative pathway for capital investment that targets carbon neutrality in China by 2060. Instead, there is some content in the discussion section that reflects the carbon peak ambition in China by 2030: *“China has promised to peak its CO₂ emissions by 2030. To achieve this target, it is projected that China's energy and CO₂ intensity levels need to decline by 43% and 45%, respectively²¹. This indicates that a substantial amount of investments in low carbon technologies are expected in the near future, and associated CO₂ emissions are mostly in the capital investment rather than the use phase.”* (Line 332–335). In addition, we added some content on the limitation related to carbon neutrality. Details are *“Lastly, shifting the fossil-based economy to a renewable energy-based economy and achieving carbon peak and neutrality will request significant capital investments in low-carbon equipment and infrastructure. Other alternative pathways of capital development are also interesting for future exploration, yet need other methods and approaches to model the entire economic and energy structures for future capital development narratives.”* (Line 573–581).

Hope the above explanations answer the reviewer's questions.

References:

21 Mi, Z. et al. Socioeconomic impact assessment of China's CO₂ emissions peak prior to 2030. *J Clean Prod* 142, 2227-2236, doi:10.1016/j.jclepro.2016.11.055 (2017).

3, Figure 2. The aggregated yearly emissions are meaningless. It is important to show how the capital formed in this year affects the emissions in another year.

We appreciate that the reviewer raised this issue and provided a contributable suggestion. Accordingly, we added a figure to illustrate annual profile of national production- and consumption-based emissions after capital re-allocation by the year when the capital-associated emissions occurred. Please see below. Key results have been explained in the main text. *“The re-allocated capital-associated CO₂ emissions (F^K) from a certain year decrease along the lifespans of assets”* (Line 247–248). This is mainly because in China, *“capital investments could account for up to 47% of its national GDP, with an annual average growth rate of 12% since 1995”* (Line 44–45). In a current year like 2017, the re-allocated F^K for production and consumption in 2017 are mostly from the past six years. In addition, *“starting with 2013, less than 1% of the F^K was allocated from CO₂ emissions embodied in capital goods invested in 1995”* (Line 200–201).

Figure S9. Annual profiles of production- and consumption-based emissions after capital re-allocation by the year when the capital-associated emissions occurred.

4, The results and discussions are too descriptive, there are no implications on how this allocation can affect climate change as the emissions are physically released in the formation year. It is also not clear how this reallocation is linked to carbon mitigation policies and climate actions.

We made the result and discussion sections more concrete in this revised version of the manuscript. Regarding the reviewer’s concern *“It is also not clear how this reallocation is linked to carbon mitigation policies and climate actions”*, we enriched our discussions and added more

content on 1) future decision making on capital investment, 2) demand-side measures, and 3) future call for a more systematic database of energy plants under emissions-trading.

1. Future decision making on capital investment.

This set of suggestions is also included in our responses to the reviewer’s comment #2. We propose a new capital development scheme, especially for those capital-intensive but climate-friendly projects such as hydropower dams. We refer to the mortgage of economic loans, which are used to purchase products in a certain year but paying back over time. When current huge emission burdens are an important concern to launch such a capital-intensive but climate-friendly project, the idea of mortgages of emission burdens—emitting currently but complementing with emission-neutrality measures throughout the payback period to offset emission burdens of the formation year—can be applied to the project. In this context, our accounting scheme is in line with carbon neutrality, but we agree that it entails a delay in carbon mitigation. But one can argue that the mitigation is to be done when the benefit is accrued rather than upfront at the point of investment. Details in the revised manuscript are *“Historically committed’ CO₂ emissions have no influence on actual CO₂ emitted each year, but provide a new scheme to assign environmental responsibilities of capital activities over time. The new accounting scheme allocates capital-associated environmental responsibilities to capital users instead of capital producers as conventionally done. Furthermore, the new accounting scheme allocates emissions of capital formation from the year of formation over capital’s entire lifetime, which relieves emission responsibility for the year of formation yet indeed entails a delay in carbon mitigation. But one can argue that the mitigation is to be done when the benefit is accrued rather than upfront at the point of investment. Learning from economic loans or mortgages—used to purchase products in a certain year but paid back over time, our new accounting scheme offers decision makers new insights into the investment plans of capital-intensive but climate-friendly projects such as infrastructure for any renewables. When current huge emission burdens are an important concern to launch such a capital-intensive but climate-friendly project, the idea of mortgages of emission burdens—emitted during construction but complemented with emission-neutrality measures throughout the payback period to offset emission burdens of the formation year—can be applied to the project. In this context, our accounting scheme is in line with the concept of carbon neutrality. Our findings suggest to decision-makers considering this inertia of capital assets in terms of historic and future ‘committed’ environmental pressures when making capital investment plans and designing capital-related policies.”* (Line 307–327).

In addition, we think our accounting method shows more comprehensive emission responsibilities of economic activities, because it captures upstream emissions of both intermediate inputs and capital inputs. The latter is usually neglected in emissions accounting. *“Current news indicates that in light of the COVID-19-induced slump in the world economy²² and the Russia-Ukraine war²³, China has further stimulated investment, mostly in the service sectors, energy, and food products”* (Line 345–347). We believe that *“the endogenization of capital is an additionally necessary step to ensure that policy makers realize the synergies and trade-offs between capital-intensive economic development and associated environmental*

burdens, to avoid any 'lock-in' effects on carbon emissions or resource requirements" (Line 347–350).

2. Demand-side measures.

Our temporal allocation method didn't change the supply chain-wide emission results of final consumption (shown as the grey areas in Figure 2a, attached below). Instead, our method takes one step further to consider the capital inputs and embodied CO₂ emissions to produce associated final consumption (the purple areas in Figure 2a). We believe our accounting method accounts more comprehensively for emission responsibilities of final-consumption activities, because it captures upstream emissions of both intermediate inputs and capital inputs. Examples are *"we find that capital-associated CO₂ emissions are mostly attributable to the production and consumption of capital-intensive service sectors (e.g., real estate services, residential services), which are usually not regarded as main CO₂ emitters"* (Line 339–341). We argue *"failing to include this capital-associated part of CO₂ emissions in the CO₂ emission accounting especially of service sectors hence strongly underestimates their impacts on the climate."* (Line 341–343). As such, *"effective demand-side measures, such as reducing the consumption of emission-intensive products (e.g., livestock), should be complemented by reducing the consumption of capital-intensive products (e.g., big houses) based on our study."* (Line 343–345).

Figure 2. Alteration to production-based emissions (PBE) and consumption-based emissions (CBE) due to capital re-allocation. In (a), the national PBE with (PBE^K represented by the orange dashed line) and without (conventional PBE represented by red solid line) the re-allocation of capital-associated CO₂ emissions (F^K), as well as the national CBE with (stacked grey, purple and light purple areas) and without (i.e., the conventional CBE represented by black solid line) the re-allocation of F^K are shown.

3. Future call for a more systematic database of energy plants under emissions-trading.

China has launched its first national emissions-trading scheme on 16 July 2021, to increase climate mitigation measures from the energy sector. Compared to CO₂ emissions embodied in capital assets used for electricity generation, direct emissions of the sector will be its main CO₂ emissions in the future. But we still found that when using the new CO₂ accounting scheme proposed in this study, there would be an average 30% reduction from conventional PBEs of energy sector in 2030. It means that the choice of accounting methods (conventional emission accounting or the new accounting scheme considering capital-associated CO₂ emissions) influences the emission results considerably, and hence influences the emissions allowances of each plant in the emissions-trading market. The necessity to conduct such a temporal allocation was fully explained in our responses to the reviewer's comment #1. As such, "*we provide suggestions for policy makers to consider capital-associated emissions in China's emissions-trading market, particularly for energy plants: 1) constructing a systematic database¹⁵ that covers the lifetime of each device (start, retired or ceased operating date date), fuel types, and generating capacities, which determine the emissions during the operating phase; 2) developing a standard accounting method to quantify capital inputs at high resolution of assets especially for power plant structures, generating devices, and transmission lines²⁶, which reflect indirect emissions related to capital inputs, that is, 'historically committed' emissions; and 3) formulating a rational and fair price mechanism for both historic and current emissions for companies to trade their emission allowances, which help take into consideration both emissions that are related to energy plants and their roles in the emissions-trading market.*" (Line 363–373)

References:

- 15 Davis, S. J. & Socolow, R. H. Commitment accounting of CO₂ emissions. *Environmental Research Letters* 9, doi:10.1088/1748-9326/9/8/084018 (2014).
- 22 Hertwich, E. G. Increased carbon footprint of materials production driven by rise in investments. *Nat Geosci* 14, 151-155, doi:10.1038/s41561-021-00690-8 (2021).
- 23 WION. A 'troubling new strategic convergence': China plans to invest more in Russian energy, commodity firms. WION Web Team (2022).
- 26 Wei, W. et al. Embodied greenhouse gas emissions from building China's large-scale power transmission infrastructure. *Nature Sustainability* 4, 739-747, doi:10.1038/s41893-021-00704-8 (2021).

5. Overall, this paper is too technical and descriptive, it is more suitable for a disciplinary journal.

We felt so sorry that the reviewer didn't give more optimistic assessments on our study. After this round revision, we believe we have understood and addressed the reviewer's comments and major concern. We hope the revisions made in this manuscript can alleviate the reviewer's concerns and come back with more optimistic reflections on our study from the reviewer.

Reviewer #2 (Remarks to the Author):

Overall Comments:

The authors have performed a detailed and thorough analysis, building on prior work to demonstrate the importance of endogenizing capital assets into MRIO models with a case study of China, and thus offer a valuable contribution. In the comments below, I have a few questions about the methodological approach. I am in support of the consumption-based accounting approach, but am not yet convinced by the modified pseudo- production-based accounting approach. My other feedback is mainly to ensure that these complex concepts are explained thoroughly and clearly for the reader throughout, reducing any ambiguity in terminology and phrasing.

The authors thank the reviewer for the thoughtful reflection of this study and sharing his/her valuable questions and comments. We also greatly appreciate the time and efforts the reviewer generously spent reviewing this manuscript. We believe we have understood and addressed the reviewer's comments and major concerns. Particularly for "***the modified pseudo- production-based accounting approach***" raised by the reviewer, we can ensure that our re-allocation method to production-based emissions (PBE) follows the logic of conventional PBE accounting. The main reasons for the confusion lie in the unique features of capital assets. Different from non-capital goods, capital assets undergo depreciation processes, which are accounted for as consumption of fixed capital in value added on the production sites. As such, emissions caused by generating value added (in our case, emission embodied in depreciated capital) should be assigned to production sites as part of PBE. Detailed explanation can be found in our responses to the reviewer's comment #11.

We have provided responses (in indented and in black) and revised texts as they appear in the revised manuscript and supplementary information (in *indented and in blue italics*) below each comment the reviewers made (in ***black italics and bold***). The line numbers in this response letter correspond to those in the "tracked-changes" version of the revised manuscript or supplementary information.

1. L25: While capital assets contribute to CO₂ emissions over their lifetime, that is due to either the use phase impacts or embodied impacts of maintenance and repair. This is different from the concepts being discussed in the paper, which instead argue that embodied CO₂ emissions ought to be allocated across the lifetime. This concept arises again in Figure 1b, with the term "Emissions of Capital Use". Consider incorporating the term "embodied" throughout.

We agree with the reviewer that CO₂ emissions of capital assets are from both the use phase and the production phase. The latter should be clarified as "embodied" CO₂ for the temporal allocation. We carefully revised the manuscript to distinguish these two parts of emissions, as suggested by the reviewer.

Examples are “*We show that allocating CO₂ emissions embodied in capital assets over time ...*” (Line 30–31), and “*As for supply chain-wide CO₂ emissions embodied in depreciated capital (F^K), ...*” (Line 133–134).

2. L48-50: Please rephrase or reorganize this sentence to more clearly distinguish capital assets from non-capital assets; the producers of any good are usually different from their users, so this is not a clear distinction.

The related Lines 101-104 are more clear, but there the phrase “for the production of final demand” is inaccurate, rather “for production to satisfy final demand”. This appears again in 112-113.

Thanks for this suggestion. We revised associated contents accordingly and do feel current expression of capital-related activities (including investment, production or formation, and use or depreciation) is clearer than our previous expression: “*First, capital assets are invested and used by economic sectors for productive purposes. Between the initial investment and production-oriented using phase, capital assets are produced by capital-producing (so-called ‘capital formation’ in national accounting) sectors.*” (Line 50–54).

In addition, to further make the differences between these capital-related activities clear to readers, we also inserted an example in the caption of Figure 1: “*An example to understand different terms of capital in (a): the transportation services sector investing in transportation equipment is regarded as ‘capital investment’; the manufacturing sector producing the transportation equipment is regarded as ‘capital formation’; the transportation services sector taking over transportation equipment for providing transportation services is regarded as ‘capital use’; during the use of transportation equipment, the annual decline of the total value of the transportation equipment is regarded as ‘capital depreciation’.*” (Line 151–156).

We changed “for the production of final demand” into “*for production to satisfy final demand*” (Line 113–114).

3. L53-57: Please consider rephrasing to reduce confusion. Many assets are partially depreciated, but the reader might infer that instead 1/3 of assets are fully depreciated and 2/3 are new. Also, while depreciation may be due to the physical effects suggested, there are other non-physical reasons (technological, cultural) for an asset becoming obsolete, and financial depreciation estimates may/not align with physical depreciation estimates.

The reviewer raised two points of confusion.

Related to the argument that “*the reader might infer that instead 1/3 of assets are fully depreciated and 2/3 are new*”, we replaced previous content. To show the readers that capital assets can exist for several years or even decades, we used a more understandable example related to energy infrastructure, instead of showing the percentage of depreciated capital in the

total investment amount. We also cited two papers as references for our statement. Details are “*Second, capital assets can exist for several years or even decades, and serve economic production throughout their lifespans. For instance, long-lived energy infrastructure has a median lifespan of 35 years¹⁵ and is expected to contribute substantial CO₂ emissions over the next decades¹⁶.*” (Line 56–58).

With regard to capital depreciation, we agree with the reviewer that there are multiple reasons for capital assets to be “depreciated”. Physical use for economic production is (the most important) one of them, while others could be technological or cultural reasons (i.e., non-physical reasons mentioned by the reviewer). We realized that the amount of depreciation of non-physical reasons are not fully available in China and has more uncertainty among different asset types and sectors. That is why we use the perpetual inventory model (PIM) to estimate monetary depreciation of assets in each year. PIM is a geometric method and a standard practice adopted by national and international statistical agencies and researchers for constructing capital consumption time series. We acknowledge this limitation and add a brief discussion in section ***Limitation and future work*** as follows: “*Data on capital consumption are more readily available than that of capital services. In addition, capital services rely on the prices of capital assets which have higher uncertainty among provinces and for different years. Capital consumption data are frequently calculated using the PIM, which has been widely accepted by national and international statistical agencies and researchers. Thus, we also conduct this analysis by relying on capital consumption to represent the use of physical capital assets.*” (Line 552–557).

Reference:

15 Davis, S. J. & Socolow, R. H. Commitment accounting of CO₂ emissions. Environmental Research Letters 9, doi:10.1088/1748-9326/9/8/084018 (2014).

16 Davis, S. J., Caldeira, K. & Matthews, H. D. Future CO₂ Emissions and Climate Change from Existing Energy Infrastructure. Science 329, 1330-1333, doi:10.1126/science.1188566 (2010).

4. 89-91: While the Methods does describe the scenarios in depth, since they are frequently referred to in prior text, it would help the reader to have a somewhat longer description here.

As suggested by the reviewer, we extended the descriptions of the two capital-oriented investment scenarios in the introduction section. Details are “*One of the capital-oriented investment pathways is developed on the principle of improving economic growth and social well-being (KES, here ‘K’ standing for capital), under which China is increasingly focusing on the role of capital assets, especially infrastructure related to transportation and communications. The other pathway is based on the principle of low-carbon development (KLC), under which China’s future capital investment are required to focus on low-carbon technologies by the electricity generation sector and end-use sectors such as transportation services.*” (Line 96–101).

5. L123: Stating "capital use and depreciated capital" is redundant, since use of capital is equivalent to capital depreciation.

We agree with the reviewer that in most cases, use of capital is equivalent to capital depreciation. In our study, compared to "use", "depreciation" of capital is exactly the process that we want to quantify and consider in the model. Based on the reviewer's suggestion, we deleted "capital use" in the main text (Line 132–133).

6. L182-184: This is an interesting paragraph. However at the start, since conventional and re-allocated PBE and CBE accounts are both divided by the same population to get per-capita results, a relative decline in the absolute results would necessarily imply a relative decline in the per-capita results.

Thanks for raising this point. According to the reviewer's comment, we revised the first sentence of this paragraph. We hope the current sentence better summarizes the key findings that are presented in the paragraph. Details are "*Changes are also observed in per-capita CO₂ emissions and inter-provincial inequality due to re-allocating capital-associated CO₂ emissions.*" (Line 202–203).

7. L240-242: Consider incorporating the concept of "carbon lock-in" (DOI 10.1146/annurev-environ-110615-085934) with the concept of future 'committed' CO₂ emissions.

We would like to thank the reviewer for suggesting this article. Actually we did want to incorporate the concept of "carbon lock-in" in the origin manuscript. We finally decided to omit this concept because we didn't want potential readers to feel confused about all these concepts (i.e., historically committed, future committed, and lock-in) that look similar but provide different insights. We briefly introduced the lock-in concept in the discussion and cited the article: "*The endogenization of capital is an additionally necessary step to ensure that policy makers realize the synergies and trade-offs between capital-intensive economic development and associated environmental burdens, to avoid any 'lock-in' effects²⁴ on carbon emissions or resource requirements and ...*" (Line 347–350).

Reference:

24 Seto, K. C. et al. Carbon Lock-In: Types, Causes, and Policy Implications. Annual Review of Environment and Resources 41, 425-452, doi:10.1146/annurev-environ-110615-085934 (2016).

8. L254-257: The result of 83% for real estate is consistent with the 77%-86% for residential housing found by Berrill et al. (2020) with a capital-endogenized model of the US (<https://doi.org/10.1111/jiec.12953>). It could be helpful to clarify for the reader that the

remaining 17% likely reflects embodied impacts of maintenance and repair along the lifespan (considered non-capital goods), rather than a possible interpretation that the residential use phase impacts are only estimated to be 17% of the life cycle carbon footprint of a residential building. The residential use phase impacts are typically accounted for separately in input-output via consumption of electricity etc.

We appreciate that the reviewer gave this suggestion and a highly relevant article to the capital topic. We added associated descriptions and cited the article in this revised version of the manuscript: *“Particularly for real estate services and residential services, ‘historically committed’ CO₂ emissions would dominate their future CO₂ emissions from both production and consumption perspectives, accounting for more than 83% of their CO₂ emissions. The remaining emissions are attributable to economic activities of real estate services in the year 2030. This figure is in line with the 77%-86% range of changes for carbon footprints of residential housing, found by Berrill et al.¹⁹, based on a capital-endogenized model of the United States.”* (Line 279–284).

Reference:

19 Berrill, P., Miller, T. R., Kondo, Y. & Hertwich, E. G. Capital in the American carbon, energy, and material footprint Journal of Industrial Ecology, doi:10.1111/jiec.12953 (2019).

Figures:

9. In Figure 1:

I like that you cleverly distinguish the time periods in the "Consumption Perspective" nodes.

I would suggest combining the redundant "Capital Investment" and "Capital Use" columns into "Capital Investment & Use".

Also, it would be easier for the reader to trace the flows if they were colored the same as the source or target nodes, rather than monochrome.

Thanks for the compliment on the figure. As for the second point, we tend to keep both capital investment and use in the figure. Reasons are 1) there are time differences between the initial investment and the use phase of capital, so we want to clearly distinguish these two phases in capital assets' lifespans; and 2) capital investment was usually mixed with “capital formation” (as emphasized in section *Monetary capital flows and embodied CO₂ emissions over time*) and capital use was usually neglected in previous input-output studies. These two reasons exactly reflect the two features of capital assets that are emphasized in the introduction section. That is, *“capital assets are invested and used by economic sectors for productive purposes”* (Line 51) and *“capital assets can exist for several years or even decades, and serve economic production throughout their lifespans”* (Line 56–57). The two features of capital assets are the core of this study. We really hope this study can make the full lifespan of capital clear to the readers. Thanks for the suggestion, anyway.

We changed the colors of monetary capital flows and embodied CO₂ flows according to the suggestion. Please see below.

Figure 1. Monetary capital flows (a) and embodied CO₂ transfers (b) across key sectors of China's capital development.

10. In Figure 2:

-The caption helps explain the series, but it would be easier to interpret if there was a symbol to distinguish the PBE and CBE without re-allocated capital, such as PBE^* and CBE^* .

-In 2b, Please clarify what the reference values are for the percent changes, as it is not readily apparent from the chart and does not read left-to-right.

Regarding the reviewer’s first point, we try not to use too many symbols and abbreviations throughout the manuscript, which will reduce the readability of the main text. Of course, on the other hand, symbols and abbreviations will make the manuscript more concise. We balanced pros and cons, and decided to use symbols “ PBE^K ” and “ CBE^K ” to represent conventional PBE and CBE after F^K re-allocation, respectively, in this revised version. If the reviewer has a second thought regarding the symbols, please let us know.

It would indeed confuse the reader if we didn’t clarify the reference values. Thank the reviewer for pointing out this issue. We revised the figure by deleting the relative changes in percentage. In addition, we also added a table in the supplementary information (Table S9), which lists all the raw data used in Figure 2. Please see below. Hope the revised version is clearer.

Figure 2. Alteration to the production-based emissions (PBEs) and consumption-based emissions (CBEs) due to capital re-allocation at the (a) national, (b) regional, and (c) provincial levels. ... In (b), the changes in regional per-capita PBEs and CBEs for the year 2017 with and without the re-allocation of F^K are plotted.

Table S9. Regional per-capita PBEs and CBEs for the year 2017 with and without the re-allocation of F^K . Unit: tonnes.

Regions	Production-based			Consumption-based			Relative Changes (CBE to PBE)
	PBE	PBE ^K	Relative Changes (PBE ^K to PBE)	CBE	CBE ^K	Relative Changes (CBE ^K to CBE)	
Beijing-Tianjin	3.9	3.3	-15%	5.8	4.1	-30%	48%
Northeast	6.1	4.0	-34%	4.5	3.3	-28%	-26%
North	6.2	3.9	-37%	5.5	3.8	-30%	-12%
Central Coast	6.3	4.9	-22%	7.2	5.4	-26%	15%
South Coast	3.8	2.7	-27%	4.0	3.0	-26%	7%
Southwest	3.7	2.5	-31%	4.2	2.7	-36%	15%
Central	4.1	3.0	-27%	5.3	3.4	-36%	30%
Northwest	11.5	7.2	-38%	7.1	5.2	-26%	-39%

Methods:

11. L156: *While I agree that endogenizing capital is indeed consistent with consumption-based accounting, I am concerned that re-allocating supply chain-wide embodied emissions of capital use is not consistent with production-based accounting principles, as mentioned briefly in L394-395. Peters & Hertwich (2008) define: "Production = GHG emissions from resident institutional units" (DOI 10.1007/s10584-007-9280-1).*

From my perspective, conventional production-based accounts offer an important and complementary perspective to capital-endogenized consumption-based accounts. The former look at the emissions where and when they occur, and the latter look at final demand drivers of those emissions. If the production-based accounts are manipulated as proposed, the perspectives start to blur, and responsibility becomes less clear. To me, the article needs to provide a stronger argument for its use.

We really appreciate that the reviewer raised this concern. When we prepared this study, we did realize that our re-allocation method to production-based emissions (PBE) may cause some confusion or deviation from the conventional definition of “production-based accounting” (Peters & Hertwich 2008). Conventional PBE of a country is defined as CO₂ occurring at the production sites within a national territory. This definition is based on the logic that emissions are assigned to the place of economic production, or more accurate, to the place of value added creation. A similar logic also applies to consumption-based emissions—emissions (accruing throughout entire value chains) are assigned to the place of final demand.

We can ensure that our re-allocation method to PBE follows the logic of conventional production-based emission assignment. The main reasons for the confusion lie in the unique features of capital assets. Different from non-capital goods, capital assets undergo depreciation processes.

We take production, purchase, and use of electricity generators (representing capital assets) and coal (representing non-capital goods) as examples. We assume that electricity generators and coal are produced in China (CHN), and then purchased by the United States (USA) for USA’s economic production. The table below shows to which country value added and CO₂ emissions of electricity generators and coal should be assigned.

	Assigning to which country?	
	Value added (notes)	CO ₂ emissions (notes)
Production of		
electricity generators	CHN (as GDP of)	CHN (as PBE of)
coals	CHN (as GDP of)	CHN (as PBE of)
Depreciation processes of		
electricity generators	USA (counted as consumption of fixed capital , as GDP of)	USA (counted as F^k , as PBE^k of)
coals	—	—
Economic production using		
electricity generators	USA (as GDP of)	USA (as PBE of)
coals	USA (as GDP of)	USA (as PBE of)

From the table, we can see capital assets undergo depreciation processes, which also generate value-added. The generated value added due to capital depreciation (i.e., consumption of fixed capital) is accounted as part of GDP of the capital purchasing country (in the above case, the USA). As such, under the logic of conventional production-based accounting, we should allocate the CO₂ emissions embodied in capital depreciation (counted as F^k of PBE^k) to the purchasing country. We also added this example in the “Supplementary Information N. The logic of temporal allocation of capital-associated CO₂ emissions to production-based emissions” to help the readers understand the reallocation method.

We hope we clearly explain the logic and why our allocation method makes sense. We also welcome the reviewer to have further discussion about this allocation method.

12. L331 - 342: This is a very interesting discussion of TIFA vs NIFA. Can you please describe how the embodied impact of investments that do not become fully functional (those that 'are completely wasted' as discussed in SI section F) are reflected in your accounting approach? Or are they insignificant?

Thank the reviewer for asking this question. We double checked the data of total investment in fixed assets (TIFA), newly increased fixed assets (NIFA), and all the statistical datasets we used. There is no specific data or percentage of *non-effective* capital assets in total TIFA or NIFA. The direct answer to the reviewer’s first question is that we didn’t consider the waste part of capital assets in our model, because they are not put into production at all. As such, the embodied

emissions of the waste capital assets are more proper to be estimated by conventional input-output modelling.

To answer the reviewer's second question, we took construction sector as an example, which accounted for 58% of total gross fixed capital formation (GFCF) of China during the period of 1995-2017 (see Line 116–117 in the main text). We obtained the construction waste data from Stadler et al. (2018) for the year 2016, and assumed all the construction waste are capital asset waste. This assumption must overestimate the amount of capital asset waste, because there are other waste generated in construction work but not counted as capital assets such as ashes. We found that construction waste accounted for 2.5% of total GFCF of construction, and 1.6% of total supply chain-wide CO₂ emissions embodied in. Considering that these figures are overestimated, we thought the waste capital assets are insignificant in total build-up capital in China.

Hope the above answers the reviewer's two questions.

Reference:

Stadler, K., Wood, R., Bulavskaya, T., Södersten, C.J., Simas, M., Schmidt, S., Usubiaga, A., Acosta-Fernández, J., Kuenen, J. and Bruckner, M. (2018) EXIOBASE 3: Developing a time series of detailed environmentally extended multi-regional input-output tables. *Journal of Industrial Ecology* 22(3), 502-515.

13. L354 - 367: *The approach to disaggregate capital investment by asset type is often a challenge when endogenizing capital assets, given the lack of detailed data. I appreciate efforts made here to make the best of the data, and calls for more detailed data in Lines 323-328 (which could be expanded). However, please describe this shortcoming in the Limitations section. For instance, Miller et al. (2019) show that using detailed capital assets versus aggregated KLEMS assets leads to a very different capital IO table (DOI 10.1111/jiec.12931).*

We would like to thank the reviewer to raise this limitation. According to the reviewer's suggestions, we expended associated content in the discussion section and the limitations.

In the discussion section, we expanded the associated content related to future call to build a more comprehensive database related to energy generators: *“Nevertheless, we provide suggestions for policy makers to consider capital-associated emissions in China's emissions-trading market, particularly for energy plants: 1) constructing a systematic database¹⁵ that covers the lifetime of each device (start, retired or ceased operating date), fuel types, and generating capacities, which determine the emissions during the operating phase; 2) developing a standard accounting method to quantify capital inputs at high resolution of assets especially for power plant structures, generating devices, and transmission lines²⁶, which reflect indirect emissions related to capital inputs, that is, the 'historically committed' emissions; and 3) formulating a rational and fair price mechanism for both historic and current emissions for companies to trade*

their emission allowances, which help take into consideration both emissions that are related to energy plants and their roles in the emissions-trading market." (Line 363–373).

We realized that the more aggregated categories of capital assets, as KLEMS did, reduce the heterogeneity among specific capital assets. This is also a main limitation of input-output modelling, which is based on macroeconomics and tends to aggregate products with different environmental properties into homogeneous sectors. We added this limitation in the revised manuscript as suggested by the reviewer: *"The limited number of categories of capital assets also raise uncertainty in asset-specific depreciation and emission properties. Miller et al.⁵² showed that using detailed capital assets versus aggregated KLEMS assets leads to very different capital-input coefficients. This limitation is in line with a main limitation of IO modelling to aggregate products with different environmental properties into homogeneous sectors. This limitation hence states the importance to build such a capital database with a high resolution of types of capital asset and investing sectors."* (Line 557–562). In addition, to address this limitation, measures could be developing a more disaggregated database covering as many assets as possible, or reducing the research area and scale to the firm-level (with finer scale data related to capital investment, depreciation, and disposal). Follow-up work focusing on project-based capital investment and associated economic-environmental-social impacts is in process, which we think will well address this limitation.

Reference:

15 Davis, S. J. & Socolow, R. H. Commitment accounting of CO2 emissions. Environmental Research Letters 9, doi:10.1088/1748-9326/9/8/084018 (2014).

26 Wei, W. et al. Embodied greenhouse gas emissions from building China's large-scale power transmission infrastructure. Nature Sustainability 4, 739-747, doi:10.1038/s41893-021-00704-8 (2021).

52 Miller, T. R. et al. Method for endogenizing capital in the United States Environmentally-Extended Input-Output model. Journal of Industrial Ecology 23, 1410-1424, doi:10.1111/jiec.12931 (2019).

14. L370: I am unclear how other countries are considered in this model. This line 370 states: "We adapted the global capital endogenized MRIO model into a Chinese inter-provincial case", while L84-87 states: "We first develop an inter-provincial capital-endogenized multi-regional input-output (MRIO)". The SI tables focus on provinces and does not list the foreign country resolution (which is important, since lower country resolution could impact accuracy).

We thank the reviewer for asking this question. Our previous expression "We adapted the global capital endogenized MRIO model into a Chinese inter-provincial case" did cause some confusion. A direct answer to the reviewer's question is we only used Chinese inter-provincial MRIO tables for this study and didn't cover other countries. This study mainly focuses on the capital

development and embodied CO₂ emissions, as well as associated trade across provinces, within China. A global analysis of Chinese capital development and its impacts related to global final consumption was recorded in our previous paper (Ye et al. *Environ. Sci. Tech.* 2021). For details please see our responses to the comment #15.

We revised the associated content in this revised version to make the scope of our model clearer. Details are “*The procedures to trace and allocate the contribution of year t’s capital investment to year n’s inter-industrial production networks follows the global capital endogenized MRIO model¹⁴. In this study, we develop a Chinese version of the capital endogenized MRIO model (details about the procedures can be found in **Supplementary Information H**).*” (Line 413–417).

Reference:

14 Ye, Q. et al. Linking the Environmental Pressures of China's Capital Development to Global Final Consumption of the Past Decades and into the Future. Environ Sci Technol 55, 6421-6429, doi:10.1021/acs.est.0c07263 (2021).

15. Relatedly, there is minimal discussion throughout the paper on China's role as a chief global exporter, and the influence that has on differences between CBE with and without capital endogenized. Assuming this is a global model, it would be useful to explain that the amount of emissions in this gap would be re-allocated for production to satisfy other countries' final demand.

We totally agree with the reviewer that the role of China in satisfying global final demand is important. As we explained in the responses to the reviewer’s comment #14, the capital endogenized MRIO model used in this study is based on the Chinese interprovincial MRIO tables, not a global model. A global analysis of Chinese capital development and its impacts related to global final consumption was recorded in our previous paper (Ye et al. *Environ. Sci. Tech.* 2021). In the previous paper, we quantify the linkages between six environmental pressures (e.g., primary energy, water, metal ores, and GHG emissions) associated with China’s capital formation attributable to global consumption, based on a global capital-endogenized MRIO model. We found that the outsourcing of capital services and the associated environmental pressures are considerable, ranging from 14 to 25% of studied environmental indicators. Without accounting for the capital–final consumption linkages, one would miscalculate China’s environmental footprints by big margins, from –61% to +114%.

Reference:

Ye, Q. et al. Linking the Environmental Pressures of China's Capital Development to Global Final Consumption of the Past Decades and into the Future. *Environ Sci Technol* 55, 6421-6429, doi:10.1021/acs.est.0c07263 (2021).

16. L388-403: I noticed "Fk allocated to GFCF" in Figure 2a. Please carefully consider if there is double-counting embedded in associated equation 3 in L402. If the GFCF is fully endogenized (and therefore removed from Y), then this approach would seem to result in the embodied impacts of that capital being accounted for twice over time.

We can ensure there is no double-counting in our results. Emissions embodied in GFCF (accounted for by a conventional IO model) are fully omitted before the re-allocation of emissions embodied in capital depreciation (F^k). Please see Line 460–463, "Two steps are taken to re-assess provincial CO₂ emissions. One is omitting the conventional PBE and CBE that are related to GFCF of a province. The other is adding back F^k re-allocated to capital using sectors generating PBE after F^k re-allocation (PBE^k), or adding back F^k re-allocated to final demand generating CBE after F^k re-allocation (CBE^k)." In addition, the re-allocation of F^k to gross fixed capital formation (GFCF) is necessary because the depreciated capital is to produce all the goods and services, part of which are capital assets.

SI:

17. Figure S1: To me it would be more appropriate to label "GFCF in final demand (%)" rather than "in Value-Added", since GFCF is a component of final demand, while CFC is a component of Value-Added.

Thank the reviewer for this suggestion. Indeed, GFCF is one category of final demand, while CFC is included in value-added. We change the label of y-axis in Figure S1. Please see below.

Figure S1. Capital investment (left y-axis) and the share of gross fixed capital formation (GFCF) in national value-added (right y-axis) of China during the period 1995-2017.

18. Figure S4: Please provide a legend for the colors.

The legend has been added in Figure S4. Please see the attached figure below.

Figure S4. Annual profiles of national carbon emissions with the re-allocation of capital-related emissions (F^K) during the period 2018-2030.

Typos (and their corrections):

19. L122: precious (previous)

Thanks for pointing out these typos. We have amended typos found by the reviewer and carefully checked the languages throughout the revised manuscript.

20 L396: constant (consistent)

We have amended this typo in the revised manuscript.

21. SI Section A: "The lack (?) of Ye et al.3 is the lack of understanding how the existing capital assets will sever (serve) future"

Here "The lack" should be "*The limitation*". We have amended it in the revised supplementary information. Thanks for this comment.

Reviewer #3 (Remarks to the Author):

This article discusses the reallocation of impacts from capital investments to the capital-demanding sectors, regions, or time-periods.

I am impressed by the amount of data and the way the authors have processed all this information into a concise manuscript.

The authors thank the reviewer for his/her positive reflections of the manuscript and sharing his/her valuable questions and comments. We also apologize, sincerely, for the exclusion of our responses to these comments in the first-round revision report, due to an unexpected cut-off issue. We believe the Editor who is handling this manuscript has specified the reasons for the exclusion of our responses and the unexpected cut-off issue to the reviewer.

In this round of revision, we mainly addressed the reviewer's concerns about the main questions that are answered in our study and how our capital-endogenized approach can benefit environmental impact assessments in both IO and LCA domains. We have provided responses (in indented and in black) and revised texts as they appear in the revised manuscript and supplementary information (in *indented and in blue italics*) below each comment the reviewers made (in ***black italics and bold***). The line numbers in this response letter correspond to those in the "tracked-changes" version of the revised manuscript or supplementary information.

1. At the same time, I am wondering why they are undertaking this exercise. Figure 1 illustrates the difference between a production and consumption perspective. I would claim that in a production perspective, the producing sector is responsible, while in a consumption perspective the consuming sector is responsible. Therefore, the impacts from capital formation are to be allocated to the capital-producing industry in a production perspective, and to the final consumer in a consumption perspective. So there would be no need to reallocate the impacts from capital investments to capital-demanding sectors. I understand the article as an attempt to create a sort of in-between thing: reallocating capital to producing industries. I may be wrong in this perception, but in any case, the article now reads in that way. Take the sentence "Our study reveals that conventional estimations of supply chain-wide CO2 emissions of 'capital investment' are, to some extent, misleading the allocation of capital-associated emission responsibilities to capital producers instead of capital users." (lines 117-119.) Words like "misleading" suggest that the an incorrect answer is given to a certain question, but that obviously depends on what the question is. If we want to know "who is emitting how much", the production perspective is clearly right, without any reallocation. If we want to know "which final demand is responsible for how much", we need to reallocate production and capital to the final use. Also lines 133-135 ("When reallocating this part of capital-associated CO2 emissions to the actual capital consumers or further to final goods and services throughout the full lifespan of capital, it would substantially alter CO2 emission accounting at both regional and sectoral levels") testify the confusion and lack of clarity about the purpose of the whole exercise.

My main concern is therefore that it is unclear which question(s) the authors have in mind.

The reviewer pointed out that specific research questions answered in our study are not clear enough. The authors really appreciate this comment. Before elaborating our responses to this comment, we want to first confirm that the reviewer agrees that environmental impact assessments of production and final consumption should take into account environmental impacts related to capital inputs. Evidence is found in the reviewer’s comment #2, which clarified that environmental impacts of capital inputs have been counted in environmental impact assessments using LCA methods. However, in the input-output (IO) modelling, capital-associated environmental impacts are improperly counted or even neglected in environmental impact assessments in most of previous studies. This problem was identified by a number of articles such as Lenzen & Treloar (2004), Chen et al. (2018), Södersten et al. (2018a, 2018b, 2020), Ye et al. (2021). To this end, it is less about the research question but more about the accuracy of carbon emission accounting based on IO modelling. We believe that taking into account capital inputs and associated carbon emissions (among regions, across sectors, and over time) provides more accurate carbon-emission results.

Our study hence develops an advanced capital-endogenized IO model and allocates capital-associated carbon emissions to the actual capital consumers or further to final goods and services throughout the full lifespan of capital. Here we want to further elaborate why we conduct this capital-endogenization exercise. In detail:

Carbon emissions can be accounted from the production and consumption perspectives, marked as production-based emissions (PBE) and consumption-based emissions (CBE), respectively. Conventional PBE of a country is defined as CO₂ occurring at the production sites within a national territory. This definition is based on the logic that emissions are assigned to the place of economic production, or more accurate, to the place of value added creation. A similar logic also applies to CBE—emissions (accruing throughout entire value chains) are assigned to the place of final demand. LCA follows the above logic and considers capital inputs and associated environmental impacts in either “cradle to gate” (i.e., from the production perspective) or “cradle to gate to grave” (i.e., from the consumption perspective) assessments. However, conventional IO analysis doesn’t follow the logic strictly.

The figure below shows capital activities related to electricity production.

Money icons created by vectorsmarket15, Electricity icons created by phatplus, Construction icons created by Pause08, Power plant icons created by Muhammad Atif, Power line icons created by Sumitsaengtong, Electricity, Phone charger and Air conditioner icons created by Fre epik - from www.flaticon.com

We have capital investment from power generation sector on power plant and power lines; capital formation by construction sector, instead of power generation sector itself, for the power plant and power lines; capital use and depreciation during electricity generation and transmission; and final users that consume electricity. The table below summarizes which step(s) of capital activities and associated CO₂ emissions are included in emissions assessments of electricity production and consumption by LCA and IO modelling. It should be noted that capital-associated emissions are different from direct-operation emissions like emissions of directly burning coals. The former is embodied in capital assets.

	Production-based emissions ("cradle to gate")	Consumption-based emissions ("cradle to gate to grave")
LCA	3 (depreciation of power plant only)	3+4
IO modelling		2+4

In this context, we think LCA is handling capital inputs more properly than IO modelling. That is, capital inputs in economic production like emissions embodied in power plants are included in the production-based accounting, while capital inputs in final consumption like emissions embodied in power plants and transmission lines are included in consumption-based accounting when using LCA. The word "*misleading*" mentioned in the main text implies this improper treatment of capital activities in conventional IO modelling. In detail, conventional IO modelling doesn't consider capital inputs in the production-based accounting at all, and treats capital formation as one category of final demand and counts associated emissions in the consumption-based emissions. This statement can also be found in Steubing et al. (2022), "*Another fundamental difference is that in monetary EEIOAs, following national accounting rules, the production of capital goods in a single year is separately recorded as a final demand category gross fixed capital formation. This means that the environmental interventions associated with the production of capital goods used for productive purposes (e.g., factory buildings, technical installations, machinery) are not included in the calculated CFs.*"

Our study addresses this key issue in conventional IO modelling by re-allocating capital-associated CO₂ emissions to the actual capital consumers or further to final goods and services throughout the full lifespan of capital. The temporal allocation of capital inputs in our study is improved, compared to the lifespan averages in most LCA analysis, based on detailed capital investment and depreciation time series. The most intuitive results are illustrated in Figure 1b in the main text, and also attached below.

Figure 1. Monetary capital flows (a) and embodied CO₂ transfers (b) across key sectors of China's capital development.

Reference:

Lenzen, M. & Treloar, G. J. Endogenising Capital A comparison of Two Methods. *J. Appl. Input-Output Anal.* 10, 1-11 (2004).

Chen, Z.-M. et al. Consumption-based greenhouse gas emissions accounting with capital stock change highlights dynamics of fast-developing countries. *Nature communications* 9, 3581 (2018).

Ye, Q. et al. Linking the Environmental Pressures of China's Capital Development to Global Final Consumption of the Past Decades and into the Future. *Environ Sci Technol* 55, 6421-6429, doi:10.1021/acs.est.0c07263 (2021).

Södersten, C.-J. H., Wood, R. & Hertwich, E. G. Endogenizing capital in MRIO models: the implications for consumption-based accounting. *Environ Sci Technol* 52, 13250-13259 (2018a).

Södersten, C.-J., Wood, R. & Wiedmann, T. The capital load of global material footprints. *Resources, Conservation and Recycling* 158, 104811 (2020).

Södersten, C.-J. H., Wood, R. & Hertwich, E. G. Environmental Impacts of Capital Formation. *Journal of Industrial Ecology* 22, 55-67, doi:10.1111/jiec.12532 (2018b).

Steubing, B., de Koning, A., Merciai, S. & Tukker, A. How do carbon footprints from LCA and EEIOA databases compare? A comparison of ecoinvent and EXIOBASE. *Journal of Industrial Ecology* 26, 1406-1422, doi:10.1111/jiec.13271 (2022).

2. Another point is that I miss the connection with life cycle assessment (LCA). In contrast to MRIO, an LCA is not based on an accounting year, and there are therefore no temporal reallocations needed, even while LCA takes a consumption perspective. If a capital good is needed that depreciated in 20 years, and every year the factory produces 50,000 products, then the per-product input of capital is 1 millionth. Good LCA databases include such information. See also DOI 10.1111/jiec.13271. I think the temporal reallocation is much more critical than the sectoral or regional reallocation. It might be worthwhile to focus more on this time aspect, and on the way it has been "solved" in LCA.

In our responses to the reviewer's comment #1, we state that, in our opinion, LCA is handling capital inputs in economic production and consumption more properly than IO modelling. We also think that the way LCA allocating capital inputs (mostly using lifespan averages) over time can be improved. As such, the connection between this study and the LCA domain is foreseen as 1) the way used in this study to allocate capital inputs over time, based on the perpetual inventory method (PIM), can be applied in LCA analysis; 2) good LCA databases can improve the construction of capital input matrices for economic production—providing more comprehensive production recipes in different economies.

1) Applying PIM in LCA analysis.

Compared to using lifespan-average inputs of capital goods in most LCA analysis, consumption of fixed capital should be estimated based on a gross capital stock and average lifespans of different types of capital assets, stated by European System of Accounts (ESA, 1995). The PIM, a standard geometric method, is hence advised to estimate gross fixed capital stock. PIM is based on a constant, age-independent rate of consumption of fixed capital (i.e., geometric rate or depreciation rate) to measure capital-stock time series. It formulates a straightforward link between capital investment, capital stock, and consumption of fixed capital:

$$S_{tE} = S_{tB} + I_t - \delta(I_t + S_{tB}) + X_t$$

where S_{tE} and S_{tB} are the end-year and beginning-of-the year net capital stocks, I_t is gross fixed capital formation in year t , $\delta(I_t + S_{tB})$ is consumption of fixed capital in year t , and X_t is other changes in volumes of the group of assets. More detailed information about the PIM method can be found in our responses to the reviewer's comment #4.

2) Good LCA databases can improve the construction of capital input matrices.

Life-cycle inventories of economic production have been highly relied on when constructing intermediate input tables. Also, the construction of capital input matrices, with capital assets in rows and economic sectors in columns, are the prerequisite to take capital inputs in IO modelling into account. The capital input matrices can provide more sensible results of capital

requirements in economic production, but is based on high-resolution capital data by asset types and sectors (as main drawback). Existing approaches to construct capital inputs matrices still rely on one-vector information of consumption of fixed capital (CFC) and gross fixed capital formation (GFCF) as proxy. It hence doesn't distinguish different types of capital assets and their usage patterns by different economic sectors at a satisfactory level. If good LCA databases can provide further information of inputs by different types of capital assets in economic production, it will make the construction of capital input matrices more comprehensive and accurate. Before that, a proper way to allocate capital input over time should be applied in LCA database, such as using the PIM.

We added some discussions in the main text (Line 558-570) to reflect this comment. Please also see below.

“Capital consumption data are frequently calculated using the PIM, which has been widely accepted by national and international statistical agencies and researchers. Thus, we also conduct this analysis by relying on capital consumption to represent the use of physical capital assets. The limited number of categories of capital assets also raise uncertainty in asset-specific depreciation and emission properties. Miller et al.⁵² showed that using detailed capital assets versus aggregated KLEMS assets leads to very different capital-input coefficients. This limitation is in line with a main limitation of IO modelling to aggregate products with different environmental properties into homogeneous sectors. This limitation hence states the importance to build such a capital database with a high resolution of types of capital asset and investing sectors. Another way to construct capital inputs in economic production is based on good life cycle inventory (LCI) databases which record inputs of key capital assets such as infrastructure, machinery, ICT, and etc.⁵³ One may argue that only lifespan-average capital inputs are recorded in most life cycle assessment. In this case, applying a proper approach in quantifying capital depreciation time series (e.g., the PIM) is a requirement in developing LCI databases.”

Reference:

Eurostat. European System of Accounts ESA 1995. (1995).

52 Miller, T. R. et al. Method for endogenizing capital in the United States Environmentally-Extended Input-Output model. Journal of Industrial Ecology 23, 1410-1424, doi:10.1111/jiec.12931 (2019).

53 Steubing, B., de Koning, A., Merciai, S. & Tukker, A. How do carbon footprints from LCA and EEIOA databases compare? A comparison of ecoinvent and EXIOBASE. Journal of Industrial Ecology 26, 1406-1422, doi:10.1111/jiec.13271 (2022).

3. In that sense, I also point to the confusion that is introduced in lines 272-278. This text starts by mentioning two topics, the second of which is the "temporal displacement of environmental responsibilities along capital's lifespan". Next the text says that "This study explores the second

issue, and demonstrates a new approach to quantify and allocate supply chain-wide capital inputs and associated CO2 emissions among sectors, across regions, and over time." So, is time alone the second topic, or is the triple {sector, region, time} the second topic?

We thank the reviewer to ask this important question. Time is alone the second topic, but, the other two dimensions {sector and region} that are covered in the first topic should also be addressed when demonstrating temporal allocation of capital.

Like the GHG Protocol, GHG emissions of a portfolio are categorized into three "scopes". Taking auto manufacturers as an example: they are directly responsible for any emissions released as part of their manufacturing process and a company's operations (Scope 1); they are also responsible for emissions produced to generate the electricity used to power their process and operations (Scope 2); in addition, they are responsible for emissions produced by companies supplying the materials used in the production of their cars (Scope 3). GHG emissions of each scope have different meanings, representing the significance of that scope in total GHG emissions of a company along the entire supply chain of its products. But, when we account for total GHG emissions of a company, we can account total emissions as Scope 1, or Scopes 1+2, or Scopes 1+2+3. We will not account total GHG emissions of a company only as Scope 2 or only as Scope 3.

It is similar in our capital-considered CO₂ emission accounting. Topic 1 and topic 2 cover different dimensions of capital activities and associated emission displacement. Each topic has its significances in the total capital-associated CO₂ emissions of a certain region. As such, we can account total capital-related CO₂ emissions of a region in line with topic 1 (considering the spatial displacement across regions and sectors), or in line with both topic 1+2 (considering the triple {sector, region, time}). We will not account for capital-associated CO₂ emissions of a region only in line with topic 2, as total capital-associated emissions of the region. Our study hence addresses the triple {sector, region, time} dimensions of capital-associated CO₂ emissions within China.

Some smaller points are the following:

4. * Line 339 mentions "the perpetual inventory method (PIM)", which is not further explained. The SI provides a reference to PIM, but some more information would be appreciated.

Thanks for the suggestion. We added a box (**Box S1**) in the **Supplementary Information F** to give a brief introduction of the perpetual inventory method. Details can also be found below.

Box S1. The Perpetual Inventory Method (PIM)

As stated by European System of Accounts (ESA, 1995)²⁵, consumption of fixed capital should be estimated based on a gross capital stock and average lifespans of different types of capital assets. The Perpetual Inventory Method (PIM) is advised to estimate gross fixed capital stock. In this Box the basic principles of the PIM are discussed.

Investment series. Full implementation of the PIM requires relatively long time series of gross fixed capital formation (GFCF), broken down by type of fixed assets and institutional sectors. Such a data set is pre-constructed for 31 provinces of China in this study, as described in this section F.

Depending on the economic structure of the country under consideration, certain types of assets may be important to be singled out in addition. For example, in some developing countries, cultivated assets such as livestock for breeding may be an important type of productive capital. In economies that are resource-rich, subsoil assets such as coal, oil or mineral reserves or non-cultivated biological resources such as natural forests may play an important role. General classifications of fixed capital applied in EU-KLMES (<https://euklems.eu/>) include transport equipment, ICT equipment, dwellings, computer software and databases.

Calculation of net capital stocks. The computational approach towards the measurement of capital depreciation and net capital stocks is by using a constant, age-independent rate of consumption of fixed capital (i.e., geometric rate or depreciation rate). This (simplified) practice dispenses from the need to specify extra parameters for a retirement profile and it permits to formulate a straightforward link between capital investment, capital stock, and consumption of fixed capital:

$$S_{tE} = S_{tB} + I_t - \delta(I_t + S_{tB}) + X_t$$

where S_{tE} and S_{tB} are the end-year and beginning-of-the year net capital stocks, I_t is gross fixed capital formation in year t , $\delta(I_t + S_{tB})$ is consumption of fixed capital in year t , and X_t is other changes in volumes of the group of assets. All variables are valued at average prices of a reference period which could be year t .

Depreciation rates. Computing the net stock above requires a rate of consumption of fixed capital, δ . Absent good information about the rates of depreciation, δ can be set by reference to other countries' depreciation rates of similar types of assets or other countries' lifespans of similar types of assets. A common way of estimating δ is the declining balance method with $\delta = R/T_a$ where T_a is the average lifespans of an asset a , and R is a parameter around 2²⁶. Because lifespans tend to be influenced by institutional and climatic conditions, it is preferable to use parameters from similar countries rather than from very different countries. EU-KLEMS and WORLDKLEMS provide detailed depreciation rate of different types of capital assets that are used by different economic sectors.

Consumption of fixed capital. Consumption of fixed capital is the amount of fixed assets used up during the period under consideration. More plain understanding of consumption of fixed capital is the deduction of gross capital stock. Reasons for the deduction are normal wear and tear and foreseeable obsolescence, including a provision for losses of fixed assets as a result of accidental damage which can be assured against. Based on the capital stock calculated by PIM, the consumption of fixed capital can be calculated as $\delta(I_t + S_{tB})$.

References:

25 Eurostat. European System of Accounts ESA 1995. (1995).

26 OECD. Measuring Capital OECD Manual 2009 Second Edition. (2009).

5. * It would be helpful if symbols (like those in line 348 and 389) would not only be described, but if also units could be provided. For instance, is I in line 348 in yuan? In yuan/yr? In another unit? This applies to all symbols, also in the SI.

We appreciate this constructive suggestion from the reviewer. It did cause some unclearness if the units of symbols are not given. We have added associated units of all the symbols used in this manuscript and the **Supplementary Information** materials. Examples are: “*Although NIFA (denoted as N , in Yuan per year) is more reasonable than TIFA to be used as capital investment (denoted as I , in Yuan per year) in PIM, ...*” (in Line 389-390), and “*Units of PBE, CBE, PBE^K, and CBE^K are tonnes*” (in Line 467).

6. * Please be a bit more precise in the graphs. Figure 2a has a vertical "Gt" and 2b has "CO2 emission per capita (tonnes)". I guess 2a is also CO2 emissions? Moreover, the lines in 2a look pretty smooth, while I think you had only 20 points (1995, 1996, 1997, etc, not anything in between). Similar remarks hold for Figure 3, as well as for the SI.

We appreciate the reviewer raised this issue.

The labels of y-axis in Figure 2 are correct. Figure 2a and Figure 2b illustrate carbon emissions at different perspectives and scales.

- In Figure 2a, we show the national production-based CO₂ emissions (PBE) and consumption-based emissions (CBE) accounted with and without the re-allocation of capital-associated CO₂ emissions (F^K). The unit is “Gt”.

In addition, reasons for the pretty smooth lines in Figure 2a may be the limited 20 points as well as the size of the figure—the horizontal dimension is compressed. We also checked the PBE of China in EXIOBASE 3 (Stadler et al. 2018) and plot the time-series PBE of China in the figure below. As we can see, the trends of PBE of China from EXIOBASE 3 and CEADs (the emission database used in our study) look similar.

- While in Figure 2b, we put our focus on the changes in per-carbon emissions with and without the re-allocation of F^K . The unit in Figure 2b hence is tonnes.

We double-checked Figure 3 in the main text and figures in the **Supplementary Information**. Figures having potential unclearness have been adjusted. For example, we added legend information in Figure S4, which was not given in the original version. Please see below.

Figure S4. Annual profiles of national carbon emissions (in Gt) with the re-allocation of capital-related emissions (F^K) during the period 2018-2030.

Reference:

Stadler, K. *et al.* EXIOBASE 3: Developing a time series of detailed environmentally extended multi-regional input-output tables. *Journal of Industrial Ecology* **22**, 502-515 (2018).

7. * Table 1 is also without units.

The unit of results in Table 1 is marked in the notes below the table, that is, "*Notes: We select ... Unit: Mt.*". To better display the unit of this table, we added the unit in the caption of Table 1, "*Table 1. Sectoral CO₂ emissions (in million tonnes) for the year 2030 under the 'business-as-usual' (BAU), capital for economy and social well-being (KES), and capital for low-carbon development (KLC) scenarios.*" (Line 293-294).

REVIEWERS' COMMENTS:

Reviewer #1 (Remarks to the Author):

The authors have addressed all my concerns and made substantial improvements.

Reviewer #2 (Remarks to the Author):

Please see attached document for comments.

Thank you to the authors for thoroughly addressing each of my comments. I would like to revisit and expand on a few of my prior comments:

3. L53-57: Please consider rephrasing to reduce confusion. Many assets are partially depreciated, but the reader might infer that instead 1/3 of assets are fully depreciated and 2/3 are new. Also, while depreciation may be due to the physical effects suggested, there are other non-physical reasons (technological, cultural) for an asset becoming obsolete, and financial depreciation estimates may/not align with physical depreciation estimates.

Thank you for addressing this point. However, the new sentence may serve to confuse readers in a different way, related to my prior comment #1. The contribution of CO₂ emissions from energy infrastructure, such as coal combustion for electricity, is a separate concept than the embodied capital asset emissions spread out over the lifespan, but they are co-mingled in this paragraph.

“For instance, long-lived energy infrastructure has a median lifespan of 35 years¹⁵ and is expected to contribute substantial CO₂ emissions over the next decades¹⁶”

Also, a small suggestion to make the caption more concise: *“the transportation services sector taking over transportation equipment for providing transportation services is regarded as ‘capital use’”* could be *“transportation services using vehicles is regarded as ‘capital use’”*.

9. In Figure 1:

I would suggest combining the redundant "Capital Investment" and "Capital Use" columns into "Capital Investment & Use".

I understand your point about the difference in time between investment and use; it is certainly the case with infrastructure and buildings, but often not the case with equipment. I would suggest providing a simple example to illustrate your logic, such as with a factory that takes years to be built before it goes into use by the investor, versus a truck that goes into use when the investment/purchase is made.

Also, it would be easier for the reader to trace the flows if they were colored the same as the source or target nodes, rather than monochrome.

The colors on the revised chart are both helpful and aesthetically pleasing, nicely done! Note that Consumption is misspelled.

11. L156: While I agree that endogenizing capital is indeed consistent with consumption-based accounting, I am concerned that re-allocating supply chain-wide embodied emissions of capital use is not consistent with production-based accounting principles, as mentioned briefly in L394-395. Peters & Hertwich (2008) define: "Production = GHG emissions from resident institutional units" (DOI 10.1007/s10584-007-9280-1).

From my perspective, conventional production-based accounts offer an important and complementary perspective to capital-endogenized consumption-based accounts. The former look at the emissions where and when they occur, and the latter look at final demand drivers of those emissions. If the production-based accounts are manipulated as proposed, the perspectives start to blur, and responsibility becomes less clear. To me, the article needs to provide a stronger argument for its use.

Thank you for providing the clarifying example, though my comment was not raised out of confusion about capital asset depreciation.

Mitigation incentives and potential policy implications

The conventional production-based accounting method offers a straightforward way to identify emissions as they take place and assign responsibility to those emitting, so that they have an incentive to take immediate and long-term action to reduce those emissions directly under their control. The concept of PBE^K is clever and innovative to explore. Whereas there may be cause for supplanting the conventional CBE with CBE^K as it is conceptually superior, I think there are unexplored risks with replacing the conventional PBE with PBE^K in climate policymaking and I am not convinced that it is conceptually superior. Therefore, I request that your proposals for including PBE^K in policy and decision-making be constrained by a call for research on how such policies and decisions would likely shift in a variety of circumstances and what the implications on meeting the climate targets might be.

In the discussion, you added in the revision:

"But one can argue that the mitigation is to be done when the benefit is accrued rather than upfront at the point of investment." This argument is not clear since a producer cannot do anything to mitigate the historic embodied emissions at the point that benefit is accrued (time of use). All they could do is plan to reduce their investment and use of future capital assets.

"Learning from economic loans or mortgages—used to purchase products in a certain year but paid back over time, our new accounting scheme offers decision makers new insights into the investment plans of capital-intensive but climate-friendly projects such as infrastructure for any renewables." There is however a likelihood to be addressed that climate-unfriendly projects would also adopt this mindset and could use it for greenwashing purposes, which would undermine the overall benefit of this approach.

Conceptual considerations

There are many inputs to production, including raw materials, energy, parts, services, and the use of capital assets. To me, it is an arbitrary threshold to say that producers should be held responsible for the embodied impacts from production of the capital assets they use, but not also the embodied impacts of production of the non-capital goods they purchase. I think it is interesting academic exercise to explore PBE^K due to the temporal implications, though.

To expand on this point, I describe another plausible accounting approach, PBE^{K+} , which allocates all of the embodied emissions (from use of capital plus all other inputs) to the producers of the goods and services sold for final consumption. It's similar to the CBE^K calculations, with a shift in responsibility from final consumer to final consumer. While I am not advocating for PBE^{K+} , I'm describing it to ask the authors to reflect on what makes PBE^K conceptually superior. I made the following table to clarify my point further (feel free to incorporate if useful, noting there is likely a more concise way to present it). It takes the perspective of an industry within a region, since these distinctions aren't visible when impacts are analyzed in aggregate across industries as in this study, and responsibility of different supply chain actors has implications for how emissions are allocated across borders in global MRIO models.

	PBE	PBE^K	PBE^{K+}	CBE^K
1. Responsibility of producers of capital assets	-Direct emissions from production	-No direct emissions -Embodied emissions from use of capital assets previously produced	-No emissions	-No emissions
2. Responsibility of producers of non-capital goods & services sold to other businesses	-Direct emissions from production	-Direct emissions from production -Embodied emissions from use of capital assets previously produced	-No emissions	-No emissions
3. Responsibility of producers of goods & services sold for final consumption	-Direct emissions from production	-Direct emissions from production -Embodied emissions from use of capital assets previously produced	-Direct emissions from production -Embodied emissions from use of capital assets previously produced -Embodied emissions of all non-capital inputs to production	-No emissions
4. Responsibility of consumers of goods & services sold for final consumption	-No emissions	-No emissions	-No emissions	-Embodied emissions of all non-capital inputs to production -Embodied emissions from producers' use of capital assets previously produced
Bulk of Responsibility	Producers causing direct emissions	Producers of non-capital goods & services	Producers of goods & services for final consumption	Final consumers

12. L331 - 342: This is a very interesting discussion of TIFA vs NIFA. Can you please describe how the embodied impact of investments that do not become fully functional (those that 'are completely wasted' as discussed in SI section F) are reflected in your accounting approach? Or are they insignificant?

Thank you for the interesting response. It may be interesting for the readers as well to add your findings to that section of the SI.

14. L370: I am unclear how other countries are considered in this model. This line 370 states: "We adapted the global capital endogenized MRIO model into a Chinese inter-provincial case", while L84-87 states: "We first develop an inter-provincial capital-endogenized multi-regional input-output (MRIO)". The SI tables focus on provinces and does not list the foreign country resolution (which is important, since lower country resolution could impact accuracy).

Please state clearly in the text that the MRIO model is for China alone and not global.

Reviewer #3 (Remarks to the Author):

I am happy with the extensive reply and reprocessing of the manuscript.

Reviewer #2

Thank you to the authors for thoroughly addressing each of my comments.

The authors thank the reviewer for the time and efforts the reviewer generously spent re-assessing this revised version of manuscript. In this second-round of revision, we mainly answer the reviewer's further questions related to advantages and disadvantages of using PBE^K in policy making. We also take suggestions from the reviewer when revising the manuscript.

We have provided responses (in indented and in black) and revised texts as they appear in the revised manuscript and supplementary information (in indented and blue italics with underline) below each comment the reviewers made (in original format). The line numbers in this response letter correspond to those in the "tracked-changes" version of the revised manuscript or supplementary information.

I would like to revisit and expand on a few of my prior comments:

3. L53-57: Please consider rephrasing to reduce confusion. Many assets are partially depreciated, but the reader might infer that instead 1/3 of assets are fully depreciated and 2/3 are new. Also, while depreciation may be due to the physical effects suggested, there are other non-physical reasons (technological, cultural) for an asset becoming obsolete, and financial depreciation estimates may/not align with physical depreciation estimates.

Thank you for addressing this point. However, the new sentence may serve to confuse readers in a different way, related to my prior comment #1. The contribution of CO₂ emissions from energy infrastructure, such as coal combustion for electricity, is a separate concept than the embodied capital asset emissions spread out over the lifespan, but they are co-mingled in this paragraph.

"For instance, long-lived energy infrastructure has a median lifespan of 35 years¹⁵ and is expected to contribute substantial CO₂ emissions over the next decades¹⁶"

Also, a small suggestion to make the caption more concise: *"the transportation services sector taking over transportation equipment for providing transportation services is regarded as 'capital use'"* could be "transportation services using vehicles is regarded as 'capital use'".

Our response:

We take the suggestions from the reviewer and revised the associated content in the main text accordingly. We deleted the sentence of energy infrastructure to avoid causing confusion to readers. In addition, we change the sentence into "the transportation services sector using vehicles is regarded as 'capital use'" (Line 658–659) as suggested by the reviewer.

9. In Figure 1:

I would suggest combining the redundant "Capital Investment" and "Capital Use" columns into "Capital Investment & Use".

I understand your point about the difference in time between investment and use; it is certainly the case with infrastructure and buildings, but often not the case with equipment. I would suggest providing a simple

example to illustrate your logic, such as with a factory that takes years to be built before it goes into use by the investor, versus a truck that goes into use when the investment/purchase is made.

Also, it would be easier for the reader to trace the flows if they were colored the same as the source or target nodes, rather than monochrome.

The colors on the revised chart are both helpful and aesthetically pleasing, nicely done! Note that Consumption is misspelled.

Our response:

We appreciate the further explanation from the reviewer regarding the figure. We are also glad that the reviewer is satisfied about our adjustment on Figure 1. To further clarify the time difference between capital investment and use, as mentioned by the reviewer, we added some text in the caption to reflect this issue. Details are "A time difference between capital investment and use should be also noted. Compared to equipment like vehicles, this time difference is more critical for infrastructure and buildings that takes years to be built before being used by the investors." (Line 662–664).

11. L156: While I agree that endogenizing capital is indeed consistent with consumption-based accounting, I am concerned that re-allocating supply chain-wide embodied emissions of capital use is not consistent with production-based accounting principles, as mentioned briefly in L394-395. Peters & Hertwich (2008) define: "Production = GHG emissions from resident institutional units" (DOI 10.1007/s10584-007-9280-1).

From my perspective, conventional production-based accounts offer an important and complementary perspective to capital-endogenized consumption-based accounts. The former look at the emissions where and when they occur, and the latter look at final demand drivers of those emissions. If the production-based accounts are manipulated as proposed, the perspectives start to blur, and responsibility becomes less clear. To me, the article needs to provide a stronger argument for its use.

Thank you for providing the clarifying example, though my comment was not raised out of confusion about capital asset depreciation.

Mitigation incentives and potential policy implications

The conventional production-based accounting method offers a straightforward way to identify emissions as they take place and assign responsibility to those emitting, so that they have an incentive to take immediate and long-term action to reduce those emissions directly under their control. The concept of PBE^K is clever and innovative to explore. Whereas there may be cause for supplanting the conventional CBE with CBE^K as it is conceptually superior, I think there are unexplored risks with replacing the conventional PBE with PBE^K in climate policymaking and I am not convinced that it is conceptually superior. Therefore, I request that your proposals for including PBE^K in policy and decision-making be constrained by a call for research on how such policies and decisions would likely shift in a variety of circumstances and what the implications on meeting the climate targets might be.

In the discussion, you added in the revision: "But one can argue that the mitigation is to be done when the benefit is accrued rather than upfront at the point of investment." This argument is not clear since a producer cannot do anything to mitigate the historic embodied emissions at the point that benefit is accrued (time of use). All they could do is plan to reduce their investment and use of future capital assets.

“Learning from economic loans or mortgages—used to purchase products in a certain year but paid back over time, our new accounting scheme offers decision makers new insights into the investment plans of capital-intensive but climate-friendly projects such as infrastructure for any renewables.” There is however a likelihood to be addressed that climate-unfriendly projects would also adopt this mindset and could use it for greenwashing purposes, which would undermine the overall benefit of this approach.

Conceptual considerations

There are many inputs to production, including raw materials, energy, parts, services, and the use of capital assets. To me, it is an arbitrary threshold to say that producers should be held responsible for the embodied impacts from production of the capital assets they use, but not also the embodied impacts of production of the non-capital goods they purchase. I think it is interesting academic exercise to explore PBE^K due to the temporal implications, though.

To expand on this point, I describe another plausible accounting approach, PBE^{K+}, which allocates all of the embodied emissions (from use of capital plus all other inputs) to the producers of the goods and services sold for final consumption. It’s similar to the CBE^K calculations, with a shift in responsibility from final consumer to final consumer. While I am not advocating for PBE^{K+}, I’m describing it to ask the authors to reflect on what makes PBE^K conceptually superior. I made the following table to clarify my point further (feel free to incorporate if useful, noting there is likely a more concise way to present it). It takes the perspective of an industry within a region, since these distinctions aren’t visible when impacts are analyzed in aggregate across industries as in this study, and responsibility of different supply chain actors has implications for how emissions are allocated across borders in global MRIO models.

	PBE	PBE ^K	PBE ^{K+}	CBE ^K
1. Responsibility of producers of capital assets	-Direct emissions from production	-No direct emissions -Embodied emissions from use of capital assets previously produced	-No emissions	-No emissions
2. Responsibility of producers of non-capital goods & services sold to other businesses	-Direct emissions from production	-Direct emissions from production -Embodied emissions from use of capital assets previously produced	-No emissions	-No emissions
3. Responsibility of producers of goods & services sold for final consumption	-Direct emissions from production	-Direct emissions from production -Embodied emissions from use of capital assets previously produced	-Direct emissions from production -Embodied emissions from use of capital assets previously produced -Embodied emissions of all non-capital inputs to production	-No emissions

4. Responsibility of consumers of goods & services sold for final consumption	-No emissions	-No emissions	-No emissions	-Embodied emissions of all non-capital inputs to production -Embodied emissions from producers' use of capital assets previously produced
Bulk Responsibility of	Producers causing direct emissions	Producers of non-capital goods & services	Producers of goods & services for final consumption	Final consumers

Our response:

We appreciate greatly for the time and all the inputs from the reviewer to help improve the capital-considered emission accounting from the concept and in practices.

Regarding the comments related to mitigation incentives and potential policy implications, we made several revisions to reflect the comments. First, there are advantages and disadvantages of replacing the conventional PBE with PBE^K in climate policy making. We think replacing the conventional PBE with PBE^K will be more relevant for services sectors, which are usually not regarded as main emitters but are capital-intensive in their production. As stated in our study, “we find that ‘historically committed’ CO₂ emissions are mostly attributable to the production and consumption of capital-intensive service sectors (Table 1), which are usually not regarded as main CO₂ emitters. Failing to include this ‘historically committed’ part of CO₂ emissions in CO₂ emission accounting especially of service sectors hence strongly underestimates their impacts on the climate” (Line 281–285). So for these capital-intensive sectors, including capital in PBE as PBE^K is a key step for their contribution to climate mitigation via less investment or greener investment. Using PBE^K is also relevant for product-specific environmental impact assessments, as most life-cycle assessment (LCA) does (Steubing et al. 2022). LCA does considers capital inputs and associated environmental impacts in either “cradle to gate” (i.e., from the production perspective) or “cradle to gate to grave” (i.e., from the consumption perspective) assessments of products. From the methodological perspective, we think LCA is handling capital inputs more properly than IO modelling. Last but not the least, this study takes China as the study area, which is still in the accumulating stage of capital investment and formation for economic and social development. When we look at economies that are already in the post-developing stage, like USA and European countries, they experienced capital and emission booms earlier than China, and their future capital investments are mostly for the maintenance of already built infrastructure and assets, instead of building new capital assets. Capital-associated emissions will take a large share in PBE^K of these economies, while their conventional PBE are generally lower than developing countries. Using PBE^K also enhance the equity of emission responsibility among developed and developing countries from the production perspective. Second, we deleted the sentence “But one can argue that the mitigation is to be done when the benefit is accrued rather than upfront at the point of investment” that may raise unclearness to the readers. Lastly, we added a sentence to raise the issue of greenwashing purposes if applying the emission mortgage for big capital project. In detail, “Learning from economic loans or mortgages (used to purchase products in a certain year but paid back over time), the idea of mortgages of emission burdens—emitted during construction but complemented with emission-neutrality measures throughout the payback period to offset emission burdens of the formation year—can reduce the impacts of the delay. This mortgage idea of emission burdens is hence in line with the concept of carbon

neutrality. In addition, mortgages of emission burdens are more relevant to investment plans of capital-intensive but climate-friendly projects such as infrastructure for any renewables, if current huge emission burdens are an important concern to launch such projects. There is also a need to avoid climate-unfriendly projects to adopt this mindset and could use it for greenwashing purposes when applying the mortgage idea of emission burdens.” (Line 254–266).

Regarding the comments related to conceptual considerations, we really appreciate that the reviewer shared his/her own thoughts on production-side emission accounting. The concept of PBE^{K+}, in our opinions, is more like the accounting scheme of LCA. That is, to estimate environmental impacts of a product or an industry (which was emphasized in the comments by the reviewer), LCA uses the full input inventories of raw materials, resources, energy, capital to quantify the “cradle to gate” (i.e., from the production perspective) or “cradle to gate to grave” (i.e., from the consumption perspective) environmental impacts. LCA can also distinguish goods and services that are sold to other businesses or final consumption. When talking about the inclusion of capital inputs in environmental impact assessment, we think LCA handles capital inputs more proper than IO modelling. In this context, there is no conventional PBE (i.e., without capital inputs) when using LCA for environmental impact assessment. Using LCA, the allocation of emission responsibilities of non-capital goods and services sold for final consumption is:

	No PBE in LCA	PBE ^K in LCA	PBE ^{K+} in LCA	CBE ^K in LCA
1. Responsibility of producers of capital assets		-No emissions	-No emissions	-No emissions
2. Responsibility of producers of non-capital goods & services sold to other businesses		-No emissions	-No emissions	-No emissions
3. Responsibility of producers of goods & services sold for final consumption		-Direct emissions from production -Embodied emissions from use of capital assets previously produced -Embodied emissions of all non-capital inputs to production	-Direct emissions from production -Embodied emissions from use of capital assets previously produced -Embodied emissions of all non-capital inputs to production	-No emissions
4. Responsibility of consumers of goods & services sold for final consumption		-No emissions	-No emissions	-Embodied emissions of all non-capital inputs to production -Embodied emissions from use of capital assets previously produced along the entire supply chain (including producers, transportation, and delivery, etc.)

The main differences lie in 1) PBE^K which also include embodied emissions of all non-capital inputs to production, i.e., PBE^K equal to PBE^{K+} when quantifying non-capital goods and services sold for final

consumption via LCA; and 2) CBE^K, which includes the embodied emissions from use of capital assets not only from the producers but from the whole supply chains (including the production, trade, transportation, delivery, etc. of products). The remaining issue using LCA is how to avoid double-accounting if we want to count emission responsibilities of a whole economy—including the production and consumption of both non-capital and capital goods and services in a same time period. The double-counting issue exists in quantifying the production of capital assets and the use the capital assets in production. Examples are to quantify the total emissions of an economy with vehicle manufacturing and transportation services, using LCA, we count the emissions of the production of vehicle and the use of vehicle in transportation services both. But the double-counting issue is not so critical if we only look at the environmental impacts of a certain product and an industry.

Reference:

Steubing, B., de Koning, A., Merciai, S. & Tukker, A. How do carbon footprints from LCA and EEIOA databases compare? A comparison of ecoinvent and EXIOBASE. *Journal of Industrial Ecology* 26, 1406-1422, doi:10.1111/jiec.13271 (2022).

12. L331 - 342: This is a very interesting discussion of TIFA vs NIFA. Can you please describe how the embodied impact of investments that do not become fully functional (those that 'are completely wasted' as discussed in SI section F) are reflected in your accounting approach? Or are they insignificant?

Thank you for the interesting response. It may be interesting for the readers as well to add your findings to that section of the SI.

Our response:

Thanks for this suggestion. We attached our responses to the supplementary information section 6. In detail, "A remaining question is how significant the "waste" part of capital assets (i.e., non-effective investment) would contribute to economic production and consumption, and further to the emissions of China. Given that there is no specific data or percentage of non-effective capital assets in total TIFA or NIFA, we rely on other data sources to estimate the contribution from waste. We take construction sector as an example, which accounted for 58% of total gross fixed capital formation (GFCF) of China during the period of 1995-2017 (see the main text). We obtain the construction waste data from Stadler et al. (2018)³¹ for the year 2016, and assume all the construction waste are capital asset waste. This assumption must overestimate the amount of capital asset waste because there are other waste generated in construction work but not counted as capital assets such as ashes. We find that construction waste accounted for 2.5% of total GFCF of construction, and 1.6% of total supply chain-wide CO₂ emissions embodied in. Considering that these figures are overestimated, we conclude the waste capital assets are insignificant in total build-up capital in China."

Reference:

31 Stadler, K. et al. EXIOBASE 3: Developing a time series of detailed environmentally extended multi-regional input-output tables. *Journal of Industrial Ecology* 22, 502-515 (2018).

14. L370: I am unclear how other countries are considered in this model. This line 370 states: "We adapted the global capital endogenized MRIO model into a Chinese inter-provincial case", while L84-87 states: "We first develop an inter-provincial capital-endogenized multi-regional input-output (MRIO)". The SI tables focus on provinces and does not list the foreign country resolution (which is important, since lower country resolution could impact accuracy).

Please state clearly in the text that the MRIO model is for China alone and not global.

Our response:

Thanks for this suggestion. We added a sentence to state clearly that the MRIO model is for China only. In detail, "This Chinese version relies on the inter-provincial MRIO tables of China and focuses on the impacts of capital development in China on the emission responsibilities of provinces across China. The international import and export linked to other countries are aggregated in the MRIO tables of China." (Line 355–358).